# Substrate colonization by an emulsion drop prior to spreading

Suraj Borkar [1] & Arun Ramachandran [1✉]

In classical wetting, the spreading of an emulsion drop on a surface is preceded by the formation of a bridge connecting the drop and the surface across the sandwiched film of the suspending medium. However, this widely accepted mechanism ignores the finite solubility of the drop phase in the medium. We present experimental evidence of a new wetting mechanism, whereby the drop dissolves in the medium, and nucleates on the surface as islands that grow with time. Island growth is predicated upon a reduction in solubility near the contact line due to attractive interactions between the drop and the surface, overcoming Ostwald ripening. Ultimately, wetting is manifested as a coalescence event between the parent drop and one of the islands, which can result in significantly large critical film heights and short hydrodynamic drainage times prior to wetting. This discovery has broad relevance in areas such as froth flotation, liquid-infused surfaces, multiphase flows and microfluidics.

¹ Department of Chemical Engineering and Applied Chemistry, University of Toronto, Toronto, ON M5S 3E5, Canada.
✉email: arun.ramchandran@utoronto.ca

When a drop of a liquid suspended in an immiscible medium approaches a surface under the action of an external force, e.g., gravity (Fig. 1a), the interstitial film of the suspending fluid drains down to thicknesses where non-hydrodynamic interactions become important[1–7]. This is followed by the formation of a bridge connecting the drop and the surface (Fig. 1b), which is a culmination of the growth of undulations at the drop-medium interface at a rate that depends on the disjoining pressure, viscosity, and the interfacial tension[8–12]. This mechanism is called spinodal dewetting. Depending on the surface properties and the three-phase contact angle, the bridge then grows radially outwards, completing the film rupture process.

In this paper, we present evidence of a new mechanism of the wetting of a surface by an emulsion drop, which proceeds by the diffusion-mediated nucleation and growth of the drop phase on the substrate, and the eventual merging of a nucleated site with the parent drop (Fig. 1c). To examine the drainage and wetting stages more carefully, these processes need to be slowed down significantly to allow the capture of their mechanistic details. To achieve this, we employed polymer melts (silicone oils) as the suspending medium for the majority of our experiments, due to their high bulk viscosities and the enhanced viscosities of polymer films[13–22] immobilized near surfaces (Fig. 1d).

In our experiments, a drop with radius $R$ ranging from 30 to 250 μm was introduced into an immiscible, suspending medium and allowed to settle under gravity toward a flat substrate. We used a microinterferometric technique, Reflection Interference Contrast Microscopy or RICM[23–26] (see Supplementary Note 1 for details), capable of resolving film thicknesses down to a few nanometers[27], to study the drainage dynamics of the film of suspending fluid formed between the drop and the flat substrate. Three drops/suspending fluid combinations were used and three types of substrates were studied, as detailed in Table 1. The fifth fluid combination was a control experiment; the suspending fluid —paraffin oil—being a small-molecule liquid, was expected to match Newtonian film drainage theory better than the silicone oils (SO). The silicone oils used in the experiments have a bulk relaxation time on the order of 1 ms, while the strain rates in the film during drainage are on the order of $1\ s^{-1}$ or lower, implying that the Weissenberg numbers are small[28] (Supplementary Notes 2 and 3). Thus, any deviation from Newtonian behavior when silicone oils are used as the continuous phase is expected to arise solely due to confinement-induced effects[14,18,19].

## Results and discussion

**Film drainage dynamics.** When the drop approaches the surface to create a film of suspending fluid in the contact zone, the drop shape in the film region is initially spherical (Fig. 2a; also see Supplementary Movie 1). A lubrication pressure is developed in the film of suspending fluid between the drop and the surface, which increases as the film drains[1]. This continues until the lubrication pressure balances the Laplace pressure[1], after which the film flattens and transitions into a dimpled configuration (Fig. 2a).

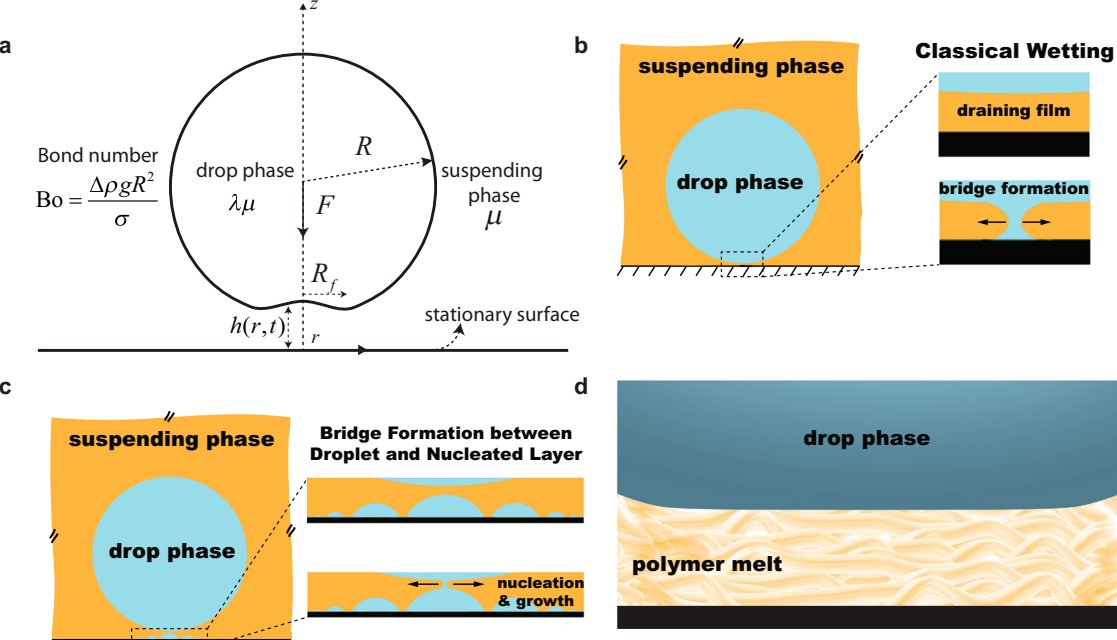

**Fig. 1 The dynamics of droplet-surface interactions in the presence of an intervening film of the suspending medium. a** Definition sketch of the film drainage process of one liquid in the approach of the drop of a second liquid toward a rigid, stationary surface. The Bond number $Bo = \Delta\rho g R^2/\sigma$ is the dimensionless number characterizing the gravitational force relative to the interfacial force acting on the drop. $\sigma$ is the interfacial tension between the drop and suspending liquids. $R_f$ is the radius of the contact zone formed due to the trapping of a film of the suspending phase between the drop and the surface. $\lambda$ is the ratio of the viscosities of the drop and the suspending phases. $F$ is the net downward external force. For a drop settling down under its own weight, the net downward force is $F = (4/3)\pi R^3 \Delta\rho g$, $\Delta\rho$ being the density difference between the drop and suspending phase, and $g$ being the acceleration due to gravity. **b** Classical spinodal dewetting theory, which involves the formation of a bridge of diameter of the order of the film thickness between the wetting phase and the substrate, followed by the growth of the bridge and subsequent film rupture. The right panel of subfigures shows a magnified version of the contact zone. **c** The new mechanism observed in our experiments shows an alternative route, whereby diffusion of the dissolved drop fluid through the film causes nucleation and growth of islands of the drop fluid on the surface. Here, a bridge between the approaching drop and one of the growing islands leads to the dewetting of the film of suspending medium, as shown in the magnified portion of the contact zone in the right panels. Note that the schematic shown is not drawn to scale. The islands are much smaller than the parent drop, and the aspect ratio depends on the island contact angle. **d** A sketch of the adsorption and layering of polymer chains over molecular length scales near a substrate, inspired by past experiments and simulations[13–22].

**Table 1 Liquid combinations and substrates employed in the experiments.**

| System | Suspending phase | Drop phase | Substrate | $\mu$ (cP)[a] | $\lambda\mu$ (cP)[b] | $\Delta\rho$ (g cm$^{-3}$)[c] | $\sigma$ (mN m$^{-1}$)[d] | $\theta$ (°)[e] |
|---|---|---|---|---|---|---|---|---|
| SO500-G-M | Silicone oil | Glycerol | Mica | 500 | 800 | 0.29 | 30 | 19 |
| SO1000-G-M | Silicone oil | Glycerol | Mica | 1000 | 800 | 0.29 | 30 | 23 |
| SO1000-G-NS | Silicone oil | Glycerol | Native SU-8 | 1000 | 800 | 0.29 | 30 | 89 |
| SO1000-G-PS | Silicone oil | Glycerol | Plasma-treated SU-8 | 1000 | 800 | 0.29 | 30 | 55 |
| PO-SO500-M | Paraffin oil | Silicone oil | Mica | 100 | 500 | 0.13 | 10 | 82 |

[a]Suspending phase viscosity.
[b]Drop phase viscosity.
[c]Density difference between the drop and the suspending phases.
[d]Interfacial tension between the drop and the suspending phases.
[e]Contact angle between the drop phase, suspending phase, and the substrate.

Further drainage makes the minimum thickness in the film comparable to separations where non-hydrodynamic interactions come into play. When these interactions are attractive in nature, the film ruptures via what has been traditionally viewed as a spinodal dewetting mechanism[8–12], and the drop spreads on the surface. When the interaction forces include a repulsive barrier, the film drainage slows down and reaches a steadystate, resulting in "black" films[29,30]. Three different temporal regimes of the minimum film thickness ($h$) can be observed in our experimental data: an initial exponential dependence ($h \sim e^{-t}$) when the film is undeformed but the pressure is less than the Laplace pressure[1] (Fig. 2b), an $h^{-1} \sim t$ dependence when the lubrication pressure approaches the Laplace pressure but the film is still undeformed (Fig. 2d), and an $h^{-3/2} \sim t$ relationship when the film drains in the dimpled configuration and has thicknesses of the order of few tens of nanometers (Fig. 2g). These behaviors are consistent with scaling analyses based on lubrication theory for a Newtonian fluid (Supplementary Note 2). Scaling theory also suggests that the transition height for either regime change should be proportional to $R^3$, and that the film radius $R_f$ in the dimpled configuration should be proportional to[1] $R^2$ (Supplementary Note 2). Our experiments agree with both trends (Fig. 2c, e, f). A recovery of Newtonian drainage theory is thus mostly observed for all three fluid combinations and for different drop radii, with either freshly cleaved mica or native SU-8 as the substrate. There are some startling exceptions, however, and these arise for the SO500-G-M and SO1000-G-M systems, particularly when $h$ begins to approach few tens of nanometers. For these systems (Figs. 2g and 3a), the drainage slows down dramatically at these thicknesses, and eventually, the minimum film thickness *increases* with time.

The retardation of drainage for small film thicknesses for SO500-G-M and SO1000-G-M is expected and is by design. It is well-known that when polymer melts or macromolecules such as silicone oil are confined between two rigid surfaces to film thicknesses of 8-10 molecular diameters, the thin film ceases to behave as a structureless continuum, and exhibits increased viscosities, higher relaxation times, slip-stick properties and other rich and complex behaviors[13–22]. Experiments performed specifically on silicone oil films down to 10 nm thicknesses have shown layering and pinning of silicone oil chains[31–33]. The reduced degrees of freedom attainable by the chains leads to a repulsive force entropic in origin[14,16,18,21] upon confinement by the glycerol drop, and the immobilized film now supports the weight of the glycerol drop to counter the tendency of the drop to approach the surface and spread on it.

**Island formation, growth, and coalescence-based wetting.** The events following the arrest of film drainage are more intriguing. An examination of the RICM images reveals the nucleation and growth of islands of a fluid phase underneath the drop (Fig. 3b and Supplementary Movie 2). Over time, these nucleated islands grow and also coalesce with each other, eventually to a size large enough to displace the parent drop upwards and against gravity, which leads to the increase in $h$. Finally, one of the islands merges with the parent drop, leading to the drop spreading over the surface (Supplementary Movie 3). The presence of these islands on the mica surface was confirmed by repeating the experiment in a silicone polymeric fluid that cured over time scales much longer than the time over which the islands were nucleated. The cured thin film was sectioned using FIB (Focused Ion Beam) milling, and imaged using cryogenic Scanning Electron Microscopy (cryo-SEM) (Fig. 3c and Supplementary Note 4). The images clearly show islands of a different phase embedded in the thin polymeric film on the mica surface.

A clue to the nature of the islands on the mica surface is offered by the observation that the islands are absent in a control experiment that has only silicone oil above the mica surface and no glycerol drop (Supplementary Note 5). More specifically, the islands are observed to nucleate only under the glycerol drop; other portions of the mica surface are featureless (Fig. 3b and Supplementary Movie 2). This leads us to hypothesize that the islands being nucleated on the mica surface are islands of glycerol formed by the dissolution and diffusion of the drop phase across the thin film to the mica substrate. Prior literature on wetting transitions in polymer blends in the vicinity of a surface also supports this hypothesis. When a polymer blend in contact with a surface is quenched from the single-phase region of the phase diagram to a temperature under the co-existence curve, the polymer that wets the surface more strongly begins to nucleate and grow on the surface as a separate phase, either in the form of a film or attached drops[34–37]. But unlike these studies, in our case, the suspending medium was not supersaturated with glycerol. Selective precipitation of one component from a mixture on a surface can also occur in the absence of supersaturation, as reported previously in a study that inferred as many as 20 monolayers of water around hydrophilic silica particles suspended in a nonpolar medium containing dissolved water[38]. This suggests that silicone oil pre-saturated with glycerol should exhibit similar nucleation and growth of islands, even in the absence of the parent glycerol drop in the vicinity of the surface[39]. Experiments that implement this test confirm the presence of islands of a new phase nucleating and growing at the mica surface in contact with silicone oil pre-saturated with glycerol (Supplementary Note 5). The diminished growth rate of the islands in these experiments is likely due to the absence of the parent glycerol drop, which serves as a constant source of glycerol at short separations from the mica surface in the original experiments.

**Effects of substrate and fluids on coalescence-based wetting.** It is instructive to assess the impacts of the type of substrate (in terms of wettability and surface roughness) and the materials composing the drop and suspending media, on the wetting

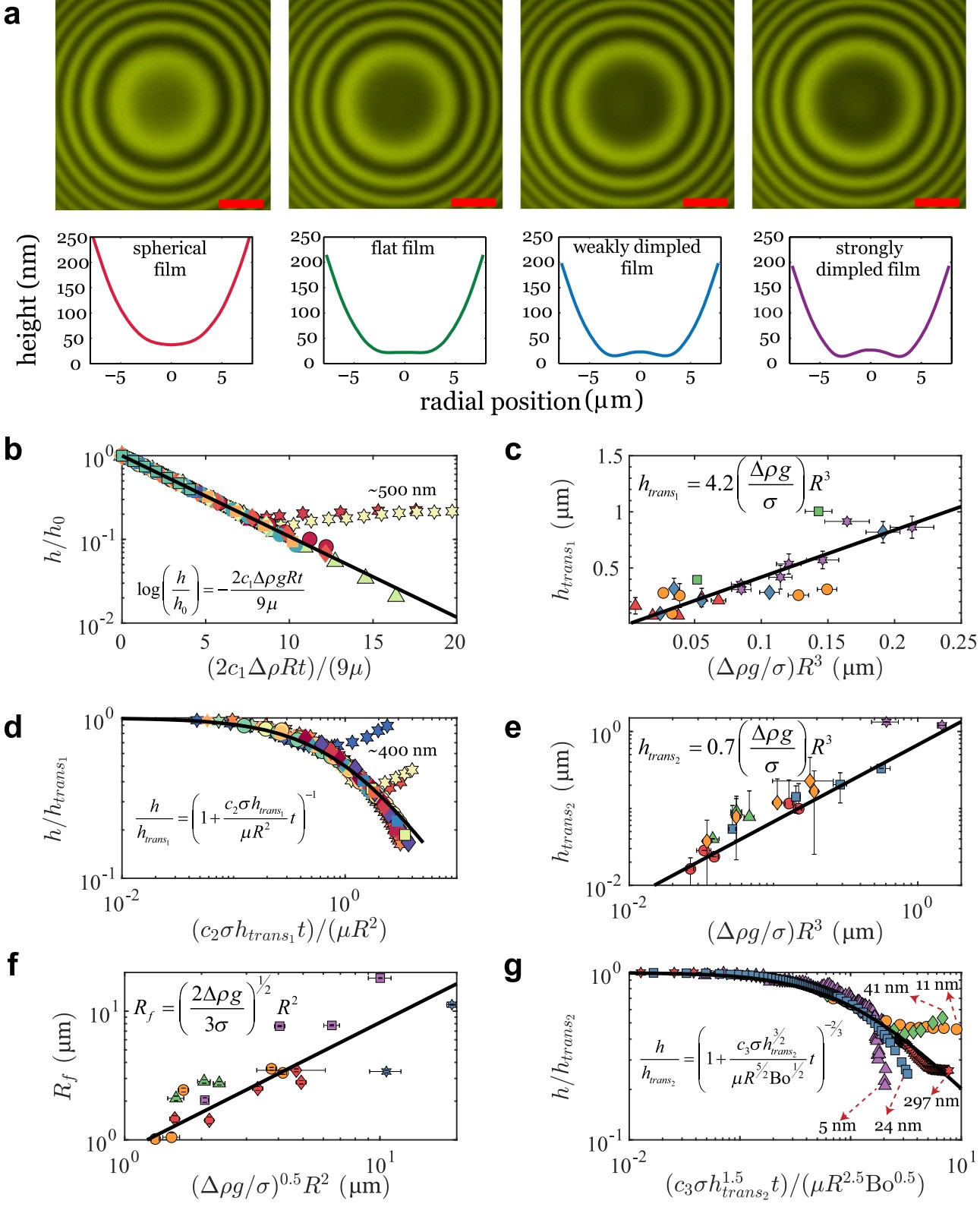

mechanism. We examined spin-coated and cured SU-8 substrates that were, after curing, used either directly (native SU-8) or after plasma treatment in the air (plasma-treated SU-8). Native SU-8, being a smooth substrate with low surface energy (SO1000-G-NS; static contact angle, $\theta \approx 89°$) relative to mica and low roughness[40] (Supplementary Note 6), did not show evidence of nucleation (Supplementary Movie 1). On the other hand, plasma treatment

of SU-8 is known to increase polar groups on the surface[40], and makes it preferentially wettable by glycerol (SO1000-G-PS; static contact angle, $\theta \approx 55°$). It also increases the roughness of the SU-8 surface significantly[40] (Supplementary Note 6). Both changes reduce the energy barrier to the nucleation of glycerol islands[41–43]. Indeed, in our experiments with plasma-treated SU-8, we observe dramatic nucleation and growth of glycerol islands

**Fig. 2 Film drainage dynamics match predictions based on Newtonian film drainage until effects of polymer confinement come into play. a** RICM images (false color; Illumination wavelength: 549 nm) and the corresponding reconstructed shapes of the drop interface with progress in film drainage. The presented result is for a glycerol drop ($R = 103\ \mu m$) in silicone oil (1000 cP), settling down toward a native SU-8 substrate. Scale bar: 5 μm (Supplementary Movie 1). **b** Exponential dependence of minimum film thickness with time when the film pressure ($P$) is less than the Laplace pressure[1] ($\sigma/R$). The agreement with theory is excellent, with the exception of the SO1000-G-PS data, for which there is a deviation and increase in the minimum film thickness with time. **c** The first transition height denoting the change from $h \sim e^{-t}$ to $h^{-1} \sim t$, is given by $h_{\mathrm{trans}_1} = A(\Delta\rho g/\sigma)R^3$, where $A$ is measured to be 4.2 ± 0.6. This transition occurs as the lubrication pressure ($P$) starts to scale as the Laplace pressure ($\sigma/R$) while the film is still spherical. **d** Minimum thickness versus time for a spherical film, when the lubrication pressure ($P$) scales as the Laplace pressure ($\sigma/R$). **e** A second transition height ($h_{\mathrm{trans}_2}$) occurs for a regime change from $h^{-1} \sim t$ to $h^{-3/2} \sim t$ where $h_{\mathrm{trans}_2} = B(\Delta\rho g/\sigma)R^3$, $B$ being measured to be 0.7 ± 0.1. This transition occurs due to a change from a spherical to a dimpled film. **f** The film radius, $R_f$, is independent of time for a constant force system (net buoyancy force here). A balance between the net buoyancy force and the force due to interfacial tension yields[1] $R_f \propto R^2$. **g** Temporal variations of the minimum film height in the dimpled regime. For the PO films, attractive forces dominate at film thicknesses of 5 nm, causing film rupture and droplet spreading. For SO1000, the film drainage appears to stop, while for SO500, the height increases with time. In **b**–**g** triangles, diamonds, circles, squares, and hexagrams correspond to data from the systems, PO-SO500-G-M, SO500-G-M, SO1000-G-M, SO1000-G-NS, and SO1000-G-PS, respectively. The different colors for each symbol are for experiments with different drop radii. The fitted values of the constants $c_1$, $c_2$, and $c_3$ for all the experiments are provided in a table in Supplementary Note 2. The error bars in **c**, **e**, **f** represent propagation errors.

underneath an approaching glycerol parent drop. The growth rates are large enough that the parent drop stops approaching the surface at a separation of a few hundred nanometers (Fig. 2b, d, g and Supplementary Movie 4).

The discovery of the coalescence-induced wetting mechanism is significant for the following reasons. Ordinarily, we treat the continuous and dispersed phases in an emulsion as immiscible, i.e., insoluble in each other; in reality, in many cases, the phases are actually sparingly soluble. And while the solubility may be low, the resistance to mass transfer across films that are only several nanometers thick can be weak. If the energy barrier for nucleation is low and, nucleation and growth kinetics are fast, then this could be the preferred mechanism for wetting. Second, the visual confirmation and characterization of the islands were possible only because we designed experiments such that the film drainage process was slow relative to the rate of nucleation and growth of the islands, by using polymeric liquids as suspending media. This allowed enough time for islands to grow to a lateral size greater than 1 μm, and could thus be visualized using RICM. In reality, the mechanism could manifest itself for small-molecule-suspending media, and the islands may need to grow only to lateral sizes of few tens of nanometers, not visible in an RICM image. Thus, coalescence-induced wetting could be the predominant mechanism of wetting. To test this idea, we performed a set of drainage experiments where a glycerol drop approached a plasma-treated SU-8 surface with a small-molecule liquid—castor oil, as the intervening suspending medium. As can be seen in the RICM images in Supplementary Movie 5, as the drop approaches the surface, the nucleation and growth processes on the substrate are so rapid that wetting takes place at a critical height of about 200 nm, a thickness over which confinement effects related to the interaction between castor oil and the substrate are expected to be negligible[13,15–17].

**Delayed coalescence between island and parent drop.** A curious observation in our experiments is that we witness extended growth periods (time scales ranging from several hours to days) of the nucleated islands that push the parent drop upwards, i.e., the islands and the parent drop do not coalesce readily. This is surprising, considering that coalescence between the parent drop and one of the islands is typically expected to be rapid for clean interfaces devoid of surfactants and in the presence of attractive van der Waals forces between like phases, indicating a repulsive barrier to coalescence. We rule out the possibility of electric double layer (EDL) forces caused due to interfacial charging at aqueous phase-silicone oil interfaces[44] by observing delayed coalescence even when experiments were conducted at the isoelectric point of the interface (pH 5) (Supplementary Note 7 and Supplementary Movie 6). Furthermore, our experiments with glycerol-dehydrated castor oil combination did not show delayed coalescence (again, see Supplementary Note 7), suggesting that the polymeric nature of the suspending medium plays a key role in the observed extended growth period of the islands. We can only speculate that the polymer chains are also immobilized at the SO-drop and SO-island interfaces. Based on the previous work[14,16,17], we do know that the presence of small-molecule impurities such as dissolved water can disrupt the formation of an immobilized layer of polymer chains and thereby erase the oscillatory force behavior at reduced separations. We see that when silicone oil has trace amounts of dissolved water (~75% of the solubility of water in silicone oil, see Supplementary Note 8; Supplementary Movie 7), the parent drop merges with a nucleated site almost instantly upon contact. Also, interferometric experiments wherein a glycerol drop approaches a flat glycerol-silicone oil (SO1000) interface show the formation of a steady film of silicone oil, whose film thickness is close to 10 nm (Supplementary Note 7). The reason for the presence of an immobilized polymer layer at the polymer-glycerol interface is not clear and needs further investigation, although we examine some possibilities in Supplementary Note 7.

**Effects of surfactants.** We were also curious as to whether this mechanism would occur in the presence of surfactants dispersed in the suspending phase, since most commercially available emulsion-based products are laden with interfacially active chemicals. Surfactants are known to adsorb on solid surfaces and a possible outcome is that the nucleation of islands would be retarded, if not completely prevented[45–48]. However, from classical nucleation theory[49,50], surfactants are also known to reduce the interfacial tension of the liquid–liquid interface, and could, thus, accelerate the nucleation phenomenon. We conducted experiments with castor oil having pre-dissolved Span 80 at two different concentrations—1.25 times the critical micelle concentration (CMC) and 4 times CMC (CMC value = 0.2 M[51]). Large, dense clusters of glycerol islands with foam-like features were observed underneath a glycerol parent drop approaching a plasma-treated SU-8 substrate (Supplementary Movies 8 and 9), suggesting that the effect of reduction of the interfacial tension is likely a stronger effect for this system of oil, water, surfactant, and substrate.

**Mechanism of coalescence-based wetting.** When cursorily inspected, the mechanism reported in this paper appears to defy the well-known Ostwald ripening phenomenon. Underlying

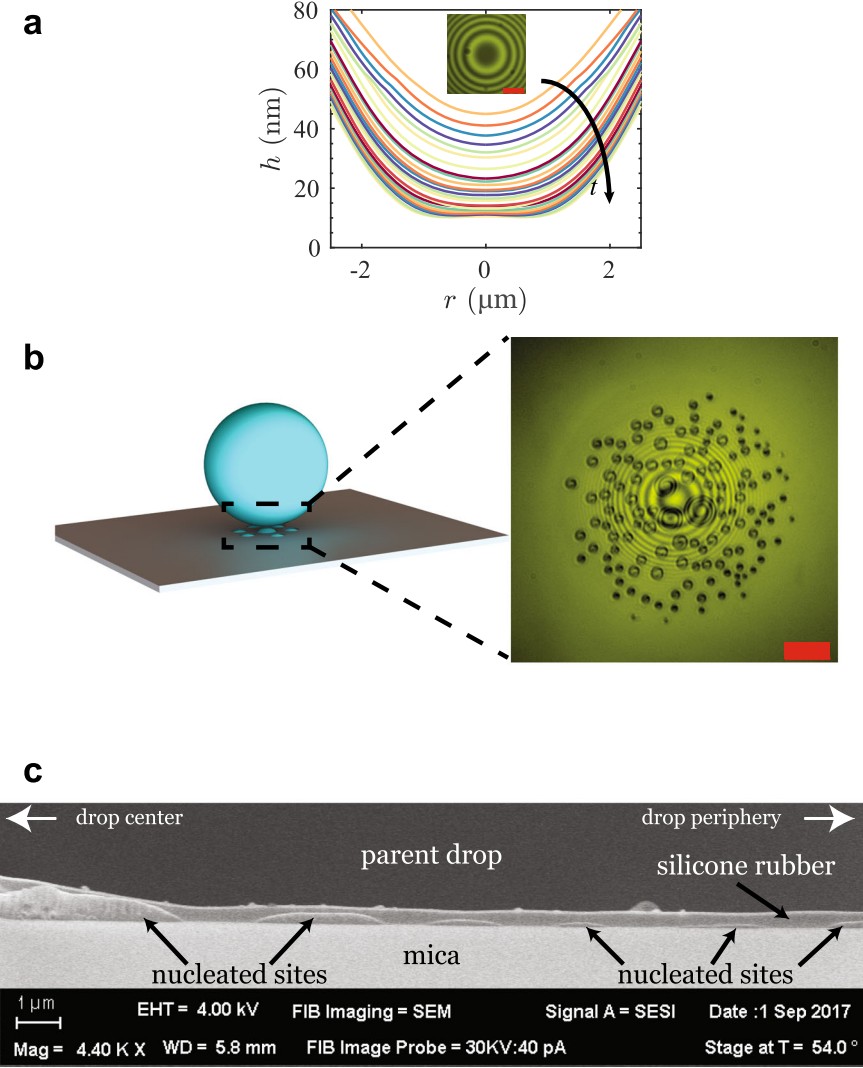

**Fig. 3 Immobilization of the silicone oil film allows for the observation of the growth of nucleated glycerol islands underneath the drop on the mica surface, to sizes capable of being resolved by the optical microscope (>~1 μm). a** Formation of a flat film of silicone oil of thickness 10 nm is observed. This solid-like immobile film, consisting of about 10 silicone oil chains (silicone oil chain width ~0.7 nm), is due to entropic repulsion upon confinement, preventing further film drainage[14,18,19]. The film shape data are for a glycerol drop ($R = 70$ μm) in SO1000 settling toward a mica substrate. Inset scale bar: 5 μm. The film drainage profiles correspond to equally spaced time stamps ($\Delta t = 10.2$ s). **b** Observation of glycerol islands underneath the drop due to transport of glycerol via diffusion, from the drop phase to the mica surface (RICM image: false color; Illumination wavelength: 549 nm). Scale bar: 10 μm. **c** Scanning Electron Microscopy (SEM) image of a Focused Ion Beam (FIB) trench showing the presence of nucleated glycerol islands at the silicone rubber/mica interface. The image was rotated by 180° to better contextualize the settling experiments. The axis of symmetry of the parent drop is located on the left edge of the SEM image while the drop periphery is situated toward the right edge. See Supplementary Note 4 for further details of sample preparation for FIB and cryo-SEM imaging.

Ostwald ripening is the Kelvin effect, according to which the solubility of a drop in a liquid is influenced by the curvature of the drop[52]. The smaller the drop radius, the larger the solubility, as conveyed by the Kelvin equation, $c_c/c_\infty = \exp(2\sigma V_m/aN_{av}KT)$, where $c_c$ is the solubility near a curved interface, $c_\infty$ is the solubility near a flat interface, $\sigma$ is the interfacial tension, $V_m$ is the molar volume of the dissolving species, $a$ is the radius of the drop, $N_{av}$ is the Avogadro number, $K$ is the Boltzmann constant and $T$ is the absolute temperature. A small drop would thereby lose mass to a neighboring larger drop by mass transfer across the suspending medium. In this work, however, we see the opposite trend—islands with high interfacial curvature are formed on a solid surface at the expense of the lower curvature parent drop. The surprising stability of the islands despite the expected loss of island material due to

curvature-enhanced solubility has been a subject of intense investigation in the nanobubble and nanodroplet literature[53]. Various mechanisms, such as interfacial mass transfer resistance[54], interfacial impurities[54], surface enhancement and dynamic equilibrium[55–58], contact line pinning[59], shielding[60,61] etc. have been considered in the literature to explain the stability. Only the mechanism of pinning of the island contact line, which prevents the curvature of the island and hence its solubility in the medium from increasing during the dissolution process, was found to be a satisfactory explanation for observing a stable, steady-state shape of the island. We examined the above possibilities for the liquids and surfaces considered in this work (Supplementary Note 9), but unfortunately, all of these mechanisms fail to explain our data. This led us to revisit an effect that has been explored in this literature

less frequently—the influence of disjoining pressure on the solubility of the island.

Consider the extended Kelvin equation that explains the effect of disjoining pressure on the solubility near a curved interface[62]:

$$\frac{c_c}{c_\infty} = \exp\left[\frac{V_m}{N_{av}KT}(\sigma\kappa + \Pi)\right]. \tag{1}$$

Here, $\kappa$ is the curvature of the drop-suspending medium interface and $\Pi$ is the disjoining pressure. As an example, consider $\Pi$ to be an attractive van der Waals interaction $\Pi = -A_H/(6\pi z^3)$, with a positive Hamaker constant $A_H$. For a film thickness $z = z_0$ comparable to $(A_H a/12\pi z)^{1/3}$, where $a$ is the radius of curvature of the drop-medium interface, the disjoining and the Laplace pressure contributions are of the same order of magnitude, and the solubility below this thickness is significantly decreased by the disjoining pressure. The lowered solubility of the drop phase in the contact line region could set up a chemical potential gradient and hence a transfer of the dissolved drop fluid from the bulk liquid to the island (see points C and D in Fig. 4a). Note that traditional Ostwald ripening with outflux from the high curvature surface of the island to the low curvature parent drop would still occur (see points E and F in Fig. 4a), implying that a net growth of the island would require the perimeter-based influx to outweigh the outflux from the surface area of the island.

**Dynamics of island growth.** To understand the physics of growth of islands, we performed long-time dynamical studies of island nucleation and growth occurring over several hours and for different systems (SO500-G-M, SO1000-G-M, and SO1000-G-PS). We implemented settling experiments lasting up to a maximum of 50 h to track the growth of the base radii of the islands underneath the glycerol drop until the point of film rupture (Fig. 4b and Supplementary Movies 3 and 4). The studies reveal two types of behaviors (Fig. 4c, d): $b^3 = b_0^3 + kt$ for SO1000-G-PS with $k$ recorded to be between 7.4 and 136 $\mu m^3$ $h^{-1}$ and $b^2 = b_0^2 + k_c t$ for SO1000-G-M and SO500-G-M with $k_c$ ranging from 0.15 to 11.9 $\mu m^2$ $h^{-1}$. Both power laws can be elucidated via the addition of glycerol at the contact line of the islands outlined above, albeit with different rate-limiting steps for mass transfer, as explained below.

Consider an overall volume balance on the island:

$$\pi\hat{V}b^2\frac{db}{dt} = \dot{v}_{IN} - \dot{v}_{OUT}. \tag{2}$$

Here, $\hat{V}(\theta) = (1 - \cos\theta)^2(2 + \cos\theta)/\sin^3\theta$ is a function that accounts for the dependence of the volume of the island on the contact angle, and $\dot{v}_{OUT}$ is the outflux from the surface area of the drop caused by the Ostwald ripening effect. The cubic dependence of the island growth rate for plasma-treated SU-8 surfaces can be explained by considering the rate-limiting step to be diffusion over a length scale of the base radius $b$ of the island (Fig. 4e, f). The influx $\dot{v}_{IN}$ ($m^3$ $s^{-1}$), is given by the scaling

$$\dot{v}_{IN} \sim \left(\frac{Dc_\infty}{b}\right)\left(\frac{2\pi bz_0}{\sin\theta}\right), \tag{3}$$

where $D$ is the diffusivity of glycerol in the suspending medium and $c_\infty$ is the bulk solubility of the drop fluid. Here, we have assumed that the island solubility is significantly reduced near the contact line, so that the driving concentration difference may be approximated as $c_\infty$. For $\dot{v}_{OUT} \ll \dot{v}_{IN}$, $b^2 db/dt \sim k$, yielding $b^3 = b_0^3 + kt$, with $k \sim 6Dz_0c_\infty/(\hat{V}\sin\theta)$. In performing the integration of the differential equation for $b$, we have assumed that the contact angle of the island is constant; this is justified in Supplementary Note 10. To estimate the diffusivity of glycerol in SO, we use the empirical power-law relationship of $D \approx 10^{-11}\mu^{-2/3}$ $m^2$ $s^{-1}$ for small molecules in

viscous oils[63], to deduce $D \approx 10^{-11}$ $m^2$ $s^{-1}$ for both SO1000 and SO500. We do not have a measure of $z_0$, the length scale of influence of the disjoining pressure on the solubility, which likely includes the effects of van der Waals attraction and hydrogen bonding. We can, however, use a scaling argument to obtain the lower bound on $z_0$ to be about 1 nm, as explained in Supplementary Note 11. The solubility of glycerol in SO was measured using NMR (Supplementary Note 12) and cross-checked with confocal Raman spectroscopy (Supplementary Note 13), to be $c_\infty = 0.02$ volume fraction units. With $\hat{V} = 0.41$ to 0.75 for $\theta = 30°-50°$, we estimate $k$ to be about 13 $\mu m^3$ $h^{-1}$. This order of magnitude prediction is in the experimentally measured range of 7.4–136 $\mu m^3$ $h^{-1}$.

In experiments where mica was used as a substrate (SO1000-G-M and SO500-G-M), we recorded a $b^2 \sim t$ behavior for the island growth. In these experiments, we expect the rate-limiting step to be diffusion across the immobilized polymer layer at the surface (Fig. 4g), arising from a significantly reduced diffusion coefficient $D_{im}$. The concentration gradient thus occurs over a length scale equal to the thickness ($d_{im}$) of the immobilized polymer layer (Fig. 4g), and hence the influx $\dot{v}_{IN}$ ($m^3$ $s^{-1}$), is given by

$$\dot{v}_{IN} \sim \left(\frac{D_{im}c_\infty}{d_{im}}\right)\left(2\pi b\frac{z_0}{\sin\theta}\right). \tag{4}$$

Again, if $\dot{v}_{OUT} \ll \dot{v}_{IN}$, $b^2 db/dt \sim b$, leading to an island growth behavior of $b^2 = b_0^2 + k_c t$, with $k_c \sim 4D_{im}c_\infty z_0/(\hat{V}d_{im}\sin\theta)$. To determine the diffusivity of glycerol in the immobilized polymer layer, we first estimate the viscosity, $\mu_{im}$ of the layer. By observing the time required for the shape of two merged islands to relax back to a circle (Supplementary Movie 10 and Supplementary Note 14), and equating it to the capillary relaxation time scale[64], we deduce $\mu_{im} \sim 10^6$ Pa s. Using, once again, the empirical relationship[63] $D \approx 10^{-11}\mu^{-2/3}$ $m^2$ $s^{-1}$, we obtain $D_{im} \approx 10^{-15}$ $m^2$ $s^{-1}$. With $\hat{V} = 0.24$ for $\theta = 17°$, and $z_0 = 1$ nm as before, we get $k_c \sim 0.4$ $\mu m^2$ $h^{-1}$, which falls in the range of the experimentally measured $k_c \approx 0.15–11.9$ $\mu m^2$ $h^{-1}$.

**Prediction of wetting time.** The knowledge of the mechanism of wetting offers the possibility of predicting the time required for a drop to wet a surface, provided that the rate-determining step or steps in the wetting process are known. As an example, let us consider the case where diffusion, nucleation, and island-drop coalescence are rapid, and the growth of the islands is the rate-controlling step. To be able to predict the wetting time scale, we need to ascertain the instant when the parent drop, settling under gravity, meets one of the islands growing on the substrate. In order for the islands to grow, they need to receive glycerol from the drop through the film of the suspending medium. Our experiments show that the islands can begin to appear at film heights as large as 1 $\mu m$. This can be explained by noting that a boundary layer of dissolved glycerol is present around the settling glycerol drop (Fig. 5a). The thickness, $\delta$, of the boundary layer can be estimated by performing a balance of the time scales of diffusion and drainage-induced convection of dissolved glycerol (Fig. 5a and Supplementary Note 15), which yields

$$\delta \sim \sqrt{\frac{\mu D}{\Delta\rho gR}}. \tag{5}$$

When the film thickness becomes comparable to the boundary layer thickness ($h \sim \delta$), the nucleation and growth of the glycerol islands commence (Fig. 5b). The variation of $h$ with time is already known, and depends on whether the film is spherical or dimpled during drainage. The island height can be deduced from its radius using the relationship, $\ell = b\tan(\theta/2)$ (Fig. 5b) and we

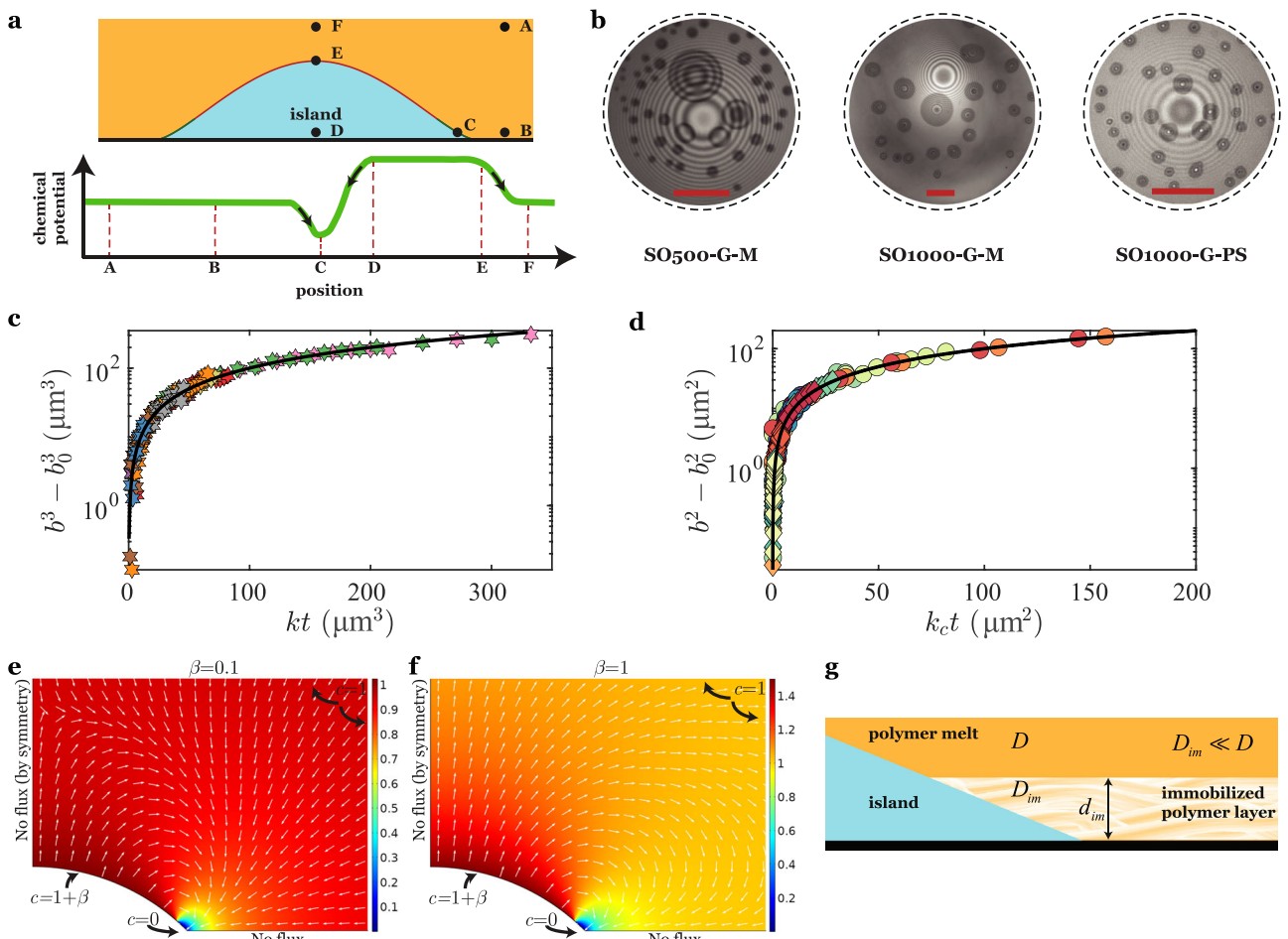

**Fig. 4 Disjoining pressure-induced modification of solubility near the contact line drives the net growth of islands. a** Schematic indicating key locations for the consideration of the chemical potential of dissolved glycerol (top), and a sketch of the variation of the chemical potential along the path $A \to B \to C \to D \to E \to F$. **b** RICM images showing the growing glycerol islands underneath a large glycerol droplet (see Supplementary Movies 3 and 4 for time lapse videos) for different cases of substrate and drop-suspending phase combinations. Scale bar: 10 μm (circular crop). **c** The growth of islands on the plasma-treated SU-8 surface follows the relationship, $b^3 = b_0^3 + kt$ (mean $k$ value of 40 μm³ h⁻¹). **d** The growth of islands on the mica surface follows the relationship, $b^2 = b_0^2 + k_c t$ (mean $k_c$ value of 1.3 μm² h⁻¹). **e, f** The Laplace equation $\nabla^2 c = 0$ was setup and solved in COMSOL, where $c$ is the concentration of the dissolved species in the suspending medium. The applied boundary conditions were: (I) $c = (1 + \beta)H(z - z_0)$ on the island surface, where $H$ is the Heaviside step function, (II) symmetry on the left edge, (III) no flux at the bottom surface owing to an impenetrable substrate, and (IV) $c = 1$ far away from the island. The parameters used in the simulation were: the contact angle, $\theta = \pi/4$, island base radius, $b = 1$, and $z_0 = 0.2 \tan(\theta/2)$. The parameter $\beta$ is the fractional increase in solubility owing to the curvature of the island. Simulations show net influx into (Fig. 4e; $\beta = 0.1$) or outflux from (Fig. 4f; $\beta = 1$) the island for weak and strong augmentations, respectively, of the solubility by the curvature of the island. The white quiver plot indicates the direction of the mass flux vector for the dissolved drop fluid. **g** An immobilized layer of silicone oil near the mica surfaces leads to a decrease in glycerol diffusivity. The rate-limiting step for mass transfer is diffusion across this immobilized layer.

already know the trend of the island radius with time. The instant, then, at which $h$ and $\ell$ become comparable, provides the critical film thickness, $h_c$, for wetting (see Supplementary Note 15 for the details of the calculation of $h_c$). In Fig. 5c, we see that $h_c$ first decreases and then increases with parent drop radius. For small drop radii, the net buoyant force on the drop is relatively weak, allowing the islands time to grow significantly and leading to spreading at film heights ranging between 100 and 300 nm. For large drop radii, the stronger gravitational body forces cause the film to transition to a dimpled shape at larger film thicknesses, thus slowing down film drainage. This leads to the critical height increasing with drop radius for large $R$. The critical heights observed in Fig. 5c are much larger than the critical heights assumed in the literature based on hydrodynamic drainage theories. In the spherical film regime, prior theories based on a

van der Waals attractive force between the drop and the surface predict that[6,7] $h_c \sim A_H^{2/5} R^{1/5}/\sigma^{2/5}$. If we consider $A_H = 10^{-19}$–$10^{-20}$ J, $R = 30$–250 μm and $\sigma = 30$ mN m⁻¹, $h_c$ lies in the range of 5–20 nm, which is significantly smaller than the experimentally recorded values (~100 nm, Fig. 5c). In the dimpled film regime, $h_c \sim A_H^{1/3} F^{1/6} R^{1/6}/\sigma^{1/2}$, where $F$ is the net gravitational force on the drop. Using the same parameters as above, $h_c$ is calculated to be between 6 and 51 nm, which is smaller than the experimental values (again, see Fig. 5c). The consequence of the underprediction of $h_c$ is a gross overprediction of the drainage time $t_D$, particularly in the dimpled regime, where $t_D \propto h_c^{-3/2}$ (Supplementary Note 2), i.e., if $h_c$ deviates by a factor of 10, the drainage time is in error by a factor of 30. Besides, in the spherical regime, $h_c$ is expected to increase with $R$, which is the opposite of our experimental observations.

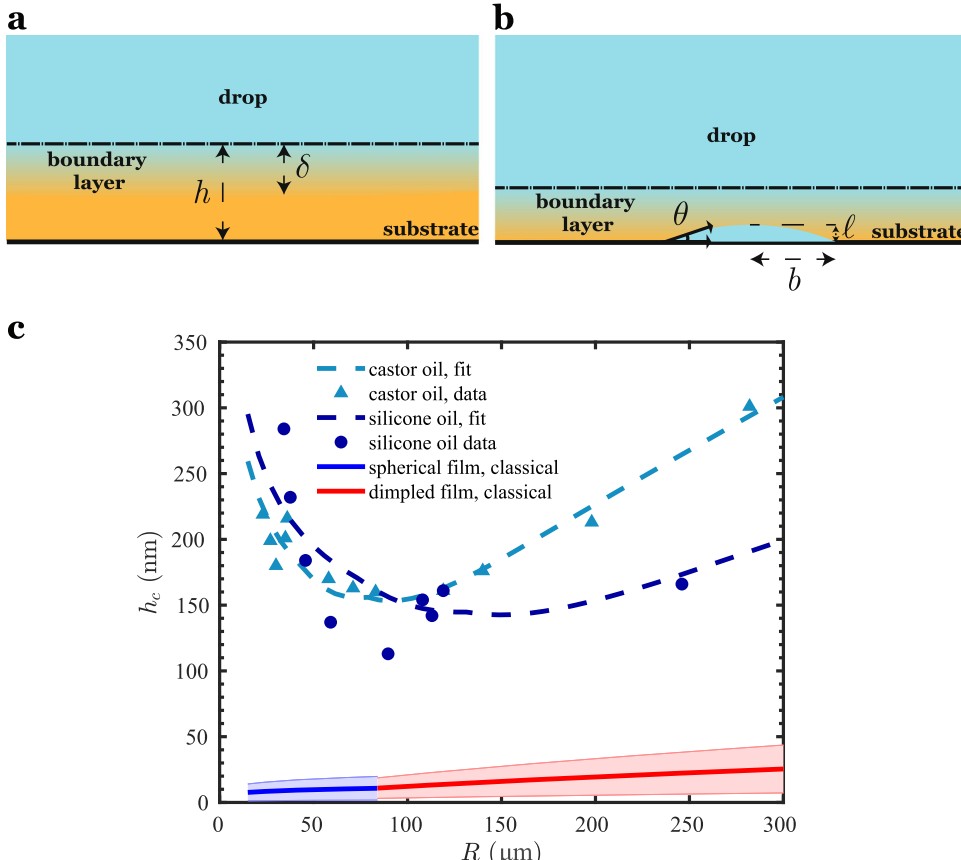

**Fig. 5 A concentration boundary layer of dissolved glycerol is present near the drop interface. a** As the parent drop settles toward the substrate, a concentration boundary layer of dissolved drop phase formed by a balance between convection and diffusion surrounds the drop-medium interface. The dash-dotted line represents the interface between the drop and the suspending fluid in the film region. The symbols $h$ and $\delta$ are the film and the concentration boundary layer thicknesses, respectively. **b** When the drop-substrate separation is nearly equal to the boundary layer thickness, the onset of nucleation takes place. The symbols $\ell$ and $b$ are the island height and radius, respectively, while $\theta$ is the contact angle. **c** The critical height, which is the film height at which the island intercepts the parent drop, as a function of the parent drop radius for glycerol drops of different sizes in castor oil and silicone oil settling toward plasma-treated SU-8. The experimental critical heights are taken to be the minimum film height before the film ruptures (castor oil), or to be the film height prior to its increase due to island growth and delayed bridge formation (silicone oil). The drainage constants (Fig. 2) were $c_1 = 1.13$, $c_2 = 0.13$ and $c_3 = 0.45$, obtained from film drainage data. The fitted values for SO1000-G-PS are $k = 8.3 \ \mu m^3 \ h^{-1}$ and $\theta = 33.8°$, which agree well with independent measurements of 7.4–136 $\mu m^3 \ h^{-1}$ and 32–55°, respectively (Fig. 4). For the castor oil/glycerol/plasma-treated SU-8 system, the constants $k = 3.6 \ \mu m^3 \ h^{-1}$, $\theta = 37.4°$, $c_1 = 1$, $c_2 = 0.1$ and $c_3 = 0.4$ were fitted. The interfacial tension ($\sigma$) used in the calculations for the two systems were measured separately to be 30 and 10 mN m$^{-1}$, respectively. The critical heights based on attractive van der Waals forces ($A_H = 10^{-20}$–$10^{-19}$ J) corresponding to spherical (blue curve) and dimpled (red curve) films were plotted taking $\sigma = 20$ mN m$^{-1}$ and $\Delta\rho = 300$ kg m$^{-3}$. The thicknesses of the shaded regions around the blue and red curves show the effect of varying $A_H$ between $10^{-20}$ and $10^{-19}$ J.

Therefore, knowledge of the mechanism of wetting is of paramount importance in order to predict the critical height and time for wetting.

In summary, we have revealed a new mechanistic picture of wetting, which is predicated on the finite solubility of the drop phase in the suspending medium. The dissolved drop fluid supplied by the parent drop is transported by diffusion to the surface to form islands, and the parent drop merges with one of the islands to accomplish wetting. Often, a key design parameter for emulsion-based coatings such as paints and inks is the time required to observe spreading once the emulsion is applied on a surface. But, while the literature is replete with studies of the ultimate outcomes of wetting (perfect, partial, or zero wetting) as governed by the spreading parameter[65], the body of work dedicated to the time it takes to achieve these outcomes is relatively sparse. This work has presented experimental evidence of the limit where the wetting time is controlled by island growth kinetics. The steps involved in the mechanism suggest that the drop solubility, the diffusivity of the drop molecule, nucleation kinetics, surface roughness, and molecular organization at the solid–liquid and liquid–liquid interfaces, may be adjusted to control the wetting time. This paper represents, therefore, a step toward a more systematic design of emulsion-based coating materials. The results in this paper are also likely to provide clues to outstanding questions related to colloidal and interfacial forces, such as the origin of the attractive forces between hydrophobic surfaces in water[27,66–68], the dynamics of the attachment of colloidal particles to liquid–liquid interfaces[69,70], and the stability of nanobubbles and nanodroplets on surfaces[53].

## Methods

**Chemicals**. Glycerol (Biotech grade) was obtained from BioShop. Silicone oil, Paraffin oil, Castor oil, and Span® 80 were obtained from Sigma Aldrich. Gel permeation chromatography (GPC) results showed the weight-averaged molecular weights $M_w$ as 16 kDa (PDI 1.73) and 20 kDa (PDI 1.49), for SO500 and SO1000, respectively.

**Optical apparatus**. A Zeiss Antiflex EC Plan-Neofluor ×64/1.25 Ph 3 objective lens mounted with adapter fittings on to a Nikon Eclipse Ti-U inverted microscope

was used to image the RICM patterns in this study. For dual-wavelength inter-ferometry, a trigger switch mechanism was used to switch between green (549 nm) and blue (485 nm) produced from a solid-state light source (SPECTRA light engine, Lumencor). The bandwidth of the light source is 10 nm. Andor Neo 5.5 sCMOS camera was used for image acquisition.

**Experimental method**. A thick block of mica (Ruby muscovite, Highest grade VI, Ted Pella Inc.) was cleaved using a razor blade down to a thickness of 10 μm. A drop of UV curable epoxy (Loctite 349) was placed on a No.1 thickness glass coverslip (VWR). The mica sheet was then placed over this and the epoxy was allowed to spread under its weight. It was ensured that no air bubbles were trapped in the epoxy. The epoxy was then cured under UV light for 15 min.

A cured PDMS rectangular cut-out was stuck around the mica sheet to form a chamber. The suspending oil was introduced into the chamber. Silica capillaries (OD: 100 μm, ID: 20 μm, Polymicro Technologies) were connected with suitable fittings to a syringe pump to introduce droplets of the desired size into the suspending oil phase.

Static contact angles were measured by placing the substrate in a rectangular cuvette (10 × 10 × 45 mm, polystyrene cuvettes, Sarstedt AG & Co.). The cuvette was filled with the suspending medium and a large glycerol ($R \sim 300$ μm) drop was introduced using a silica capillary. The drop was allowed to settle toward the substrate and eventually spread on the substrate, after which images were taken using a camera and a ×4 objective lens. These images were then used to measure the contact angle.

## Data availability
The data that support the findings of this study are available from the corresponding author upon request.

## Code availability
RICM height reconstruction in this work was performed using MATLAB. All related codes can be built with the instructions provided in Supplementary Note 1.

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

## Acknowledgements

We acknowledge the funding support from the NSERC Discovery program (Grant #402005), Discovery Accelerator Supplement (RGPAS-2018-522652), Canada Research Chair (File # 950-231567), the Ontario Early Researcher Award (Grant #ER13-09-138), Canadian Foundation for Innovation (Leaders Opportunity Fund Grant 28016) and the Ontario Research Fund. We are grateful to Prof. Janet Elliot (University of Alberta), Prof. Eugenia Kumacheva (University of Toronto), Prof. Christopher Yip (University of Toronto) and Prof. Edgar Acosta (University of Toronto) for insightful discussions. We also thank Dr. Laura Poloni and Prof. Yip at the University of Toronto for technical support with surface roughness measurements using the AFM, and Travis Casagrande and Canada Centre for Electron Microscopy (CCEM) for cryo-SEM/FIB imaging.

## Author contributions

S.B. and A.R. designed the experiments. S.B. performed the experiments, analyzed the data, and produced the figures and tables. S.B. and A.R. interpreted the data and wrote the manuscript together.

## Competing interests

The authors declare no competing interests.
