## [Peer Review File · Nature Communications]

Substrate colonization by an emulsion drop prior to spreadingReviewers' comments:

Reviewer #1 (Remarks to the Author):

Report on the manuscript entitled

Substrate colonization by an emulsion drop prior to spreading

by Suraj Borkar and Arun Ramachandran

The authors discuss in detail the mechanism how emulsion drops can merge with substrates through nucleation and growth of nanodrops on the substrate. The mechanism is new and might be suitable for publication in Nature Communications. The experiments seem to be carefully conducted. However, at the present state, the manuscript seems to be written in a hurry. A couple of important questions stay unanswered.

The mechanism put forward by the authors has some similarities to Ostwald ripening. However, in contrast to Ostwald ripening, here the small drops grow at the expense of the large drop. This topic has been discussed in the study of nanodrops on solid surfaces. There is a huge literature in the properties of nanodrops in various environment. The stability of the observed nanodrops has to be discussed in the perspective of Ostwald ripening and the existing literature on nanodrops.

The model system used in the manuscript is highly specific, because it contains no surfactant. Most emulsions contain surfactants to stabilize the emulsion drops. How would the mechanism work in the presence of surfactants that compete for the adsorption on the solid surface? This question has to be addressed.

Throughout the manuscript, the authors argue that the van der Waals interaction is the dominating interaction. How can the authors exclude other interactions, e.g. electrostatics? Due to the larger difference in dielectric constants between the oil and the glycerin drops, charges might accumulate at the (nano) drop surface. The water content also will change the Debye screening length (compare discussion on page 6). This might give alternative interpretations of the effect of water content in the sample. To clarify these points data on the ion concentration in the sample is needed, e.g. dielectric spectroscopy or conductivity measurements.

On page 6, the authors argue about an immobilized layer on the nanodrops ("We can only speculate that ..."). To me this seem very speculative and needs a solid argument to support.

Some major statement of the modelling part is unclear to me.

- The authors mention that diffusion is fast and not rate limiting. However, most of the arguments they give later on depend on diffusion (the boundary layer thickness, the growth of the nanodrop). Clarify.
- Why is the volume growth rate of the drop (Eq. 2) proportional to the circumference of the nanodroplet?
- The paragraph in the middle of page 9 (The experimentally measured critical height ...) is rather vague and unclear. Be more specific.

Minor points:

- The used silicone oil or is poorly characterized? Molar mass distribution? Purity?
- On page 5 and 6 various contact angles are reported. Under which conditions are these contact angles measured? Drop phase? Continuum phase? Advancing or receding contact angle? Hysteresis?
- In the figures the font size varies strongly. The figures should be revised to have less than a factor of two between the largest and the smallest font (including sub- or superscripts).

In summary, the manuscript needs major revisions before it might be suitable for publication in Nature Communications.

Substrate colonization by an emulsion drop prior to spreading

Review

November 15, 2018

Here is my review for the paper entitled 'Substrate colonization by an emulsion drop prior to spreading' by Suraj Borkar and Arun Ramachandran submitted for publication to Nature Communications.

Paper summary

In this paper, the authors report a study coupling experiments and scaling analysis of the transfer of liquid from the droplets of an emulsion of a polar liquid in an oil to a solid wall in close proximity and the modification of the wetting process that ensues.

The authors track the motion of a glycerol droplet in a bath of silicone oil as it approaches a wall made of mica or a resin known as SU-8. They use interferometric techniques to measure the drainage dynamics and the shape of the oil film separating the droplet from the wall. They show that the use of carefully chosen combinations of liquids and solids allow them to slow down drainage dynamics (by resorting to the confinement of polymer chains in silicone oils close to a wall, Fig. 2 and 3) enough to visualize the nucleation of domains on the solid surface whose nature is clearly different from that of the oil.

Then, the authors assume that the nucleation of these domains results from the ability of glycerol, which is *a priori* immiscible in silicone oil, to diffuse from the droplet to the wall, in a fashion that reminded me of Ostwald ripening between the droplets in an emulsion. In particular, they demonstrate that domains grow below glycerol droplets and also if silicone oil has been saturated with glycerol beforehand. They also show that the surface state of the solid and its nature control the occurrence of nucleation. From this they conclude that, given the emulsion they use, the surface of the solid must have a rather high energy and be rough enough to promote nucleation of glycerol domains.

Finally, the authors discuss the influence of the growth of these domains on wetting. In particular, they investigate the conditions under which the main droplet will coalesce

with wall domains. They show that coalescence occurs for a critical oil film thickness, that results from the balance between film drainage and island growth.

Appreciation

I have found the paper very interesting. The wetting mechanism reported by the authors is new to me, and I could not find any clue of prior observations of this wetting mechanism in the literature. The protocol used by the authors is sound. Moreover, the authors have tested rather thoroughly the dependence of the conditions under which island nucleation and growth on experimental parameters.

I think that this paper will be of great interest to many Nature Communications readers, as I can see this topic being of interest to chemical engineers, surface scientists, material scientists among many others. Besides, this work also brings up new questions with respect to the physics of solid wetting that will attract theoreticians.

I have mostly minor issues with the paper, the most important comment having to do with the derivation of the scaling laws and its comparison to Yiantsios and Davis's results. I will support publication in Nature Communications once these issues are addressed.

Minor comments

Comment 1

I have found Fig. 2 to be a bit dense, and I think that the paper would gain from improvements here. First I would recommend that the notations be clarified. For example, \hat{h} is non-dimensionalized while \hat{R} is not. Moreover, the figure would benefit from a decrease in the information content of each subfigure. For example, provided notations are clarified, *hath* could be defined in Fig. 2b and its definition won't have to be repeated in 2d and 2f.

Comment 2

Legends should also be used to tighten the style of figures. For example, in Fig. 2b, it would be great to have a box stating that the black line is a fit of an exponential function to the data. The same holds for the other figures where fits are used.

Comment 3

Figure 2 is not self-consistent, *i. e.* reading the figure with its caption still leads to confusion and the need to refer either to the body of the text or, worse in my opinion, to the supplementary materials. I would suggest that the authors define all the quantities

that are displayed in the figure (and their physical meaning) in the caption. For example, h_{trans} is not really well defined (and I find this true also in the text). I think that it is the thickness of the film for which the droplet switches from spherical to non-spherical. Is this correct? Can the authors clarify this? Finally, this leads me to not understand the difference between the data reported in 2b and 2d, reading the caption. Can the authors clarify this too? It seems to me that the distinction has to do with relative intensity of the lubrication and capillary pressure, but the way it does requires clarification.

Comment 4

I wonder about the dimensions of t^* in Fig. 2d. Given the equations in the figure, and especially the definition of h^{-1} , t^* has the dimensions of the inverse of a length, and therefore it is dimensionalized. I understand that this definition is used to test directly the dependence of the thickness on time, but the definition of t^* with no further commentary may somehow mislead the reader in thinking that this is a non-dimensionalized quantity. The same remark holds for \hat{t} in fig. 2f.

Comment 5

With respect to the fits used in Fig. 2, the authors say nothing about the prefactors that they found. Are these prefactors of order 1?

Comment 6

The authors suggest that nucleation should start once the oil film thickness is comparable to the thickness of the glycerol boundary layer in the vicinity of the droplet. I might have missed the test of this claim in the paper. Can the authors discuss this?

Comment 7

In the supplementary materials, I am not sure about the prefactors in eq. S2.7. My own calculation gives me:

$$-\partial_t h \simeq \frac{h}{a} \left[\frac{Gh^2}{12\mu} + \frac{U}{2} \right]$$

Directly integrating this equation with the condition that $h(t = 0) = h_0$ leads me to an exponential with a different argument. Where does the difference lie? Yiantsios and Davis proceed with this equation by applying asymptotics to it. Did the authors do this too? Note that integrating the equation above with the definitions of P and a given in Table 1 leads me to the same scalings as the authors.

Besides this discrepancy, I am able to find the solutions of the authors only if I assume that the Couette part of the flow is weak compared to the Poiseuille part. This assumption is not necessarily obvious. Can the authors comment on this? My understanding, given the systems they use is that they have a viscosity ratio of order 1, and it is not obvious to me that the Couette component can be neglected. A few words about this, maybe referring to Yiantsios and Davis who discuss the impact of the viscosity ratio, may be useful and instructive to the reader.

Reviewer #3 (Remarks to the Author):

In this paper, the authors investigate what happens when a liquid drop suspended in an 'immiscible' fluid approaches, because of the gravity, surfaces with different wettability properties until wetting of the drop phase on the surface occurs. The results show that the wetting occurs between the main drop and the substrate once new small droplets of the same liquid form (nucleate?) on the surface. The formation of these small droplets at the surface is attributed to the diffusion of the molecule of the liquid drop into the 'immiscible' fluid film between the drop and the substrate, followed by their nucleation at the surface.

Although the results do show some funny behavior, I have severe doubts on its originality, thus, I do not recommend for publication in Nature Communications. Some reasons are given below:

-The diffusion of liquid from drops into an immiscible fluid and its impact on its surrounding has been shown many times in different applications/fields such as microfluidics, water evaporation and emulsions. The authors can have also a look at the following papers: Su et al, Langmuir 2013,29,13339, Shimizu & Tanaka, ncomms 6,2015, 7406, Carrier et al. J.Fluid Mechanics, 2016, 798,

-The authors pretend that the results can give insights for the design of emulsion based coating materials. However the system studied in the manuscript is surfactant free and moreover the size of their drops is much larger (100 microns) than what one usually finds in emulsions (typically micron size). This will change the capillary pressure in the drops by a factor of 100, and can consequently change all the behaviour of the system.

- The authors claim that immobilization of the polymeric film between the substrate and drop leads to a highly viscous film with reduced degrees of freedom for the chains that can be structured parallel to the substrate. If this is the case I suppose that locally the diffusion coefficient should become very small especially if we will be in the presence of an anisotropic film. So, why then will diffusion be faster in this regime compared to the unconfined surrounding of the drop, in a way to induce the 'colonization'? In a similar situation, confined crystal growth experiments close to a substrate separated by a thin film of solution (Royne et al., J.Cryst.Growth, 86, 349, 2012, Cagliardi, Phys.Rev E, 97,012802 2018), it has been shown that the diffusion of ions in this thin film is rather very slow compared to the bulk and consequently this leads to some peculiar crystal shapes. Such arguments should also be valid in the experiments described here.

-Figure 3c shows larger islands at the extremities of the parent drop compared to the centre whereas figures 3b and 4 and the movies show a more homogeneous distribution. Is there any explanation for this observation?

-Is there any critical length for the 'nucleation' of the islands? Again in classical nucleation and growth such a critical length is present (see for example J. Aizenberg et al, Nature 398, 1999). In addition, It could well be that first a microscopic film forms on the surface, which subsequently destabilizes into the droplet configuration.

Reviewer #4 (Remarks to the Author):

It indeed is a very good work that deserves to be published in Nature Comm. I have only one concern which is about the uncertainties that are not provided on Figs. 2c, 2e.

RESPONSES TO THE COMMENTS AND QUESTIONS OF REVIEWER#1

We would first like to thank the reviewer for the numerous helpful comments and questions that have helped to greatly increase the clarity and content quality of our manuscript. Our detailed responses are presented below.

- 1. The mechanism put forward by the authors has some similarities to Ostwald ripening. However, in contrast to Ostwald ripening, here the small drops grow at the expense of the large drop. This topic has been discussed in the study of nanodrops on solid surfaces. There is a huge literature in the properties of nanodrops in various environment. The stability of the observed nanodrops has to be discussed in the perspective of Ostwald ripening and the existing literature on nanodrops.**

Response:

We thank the reviewer for raising this point. We have now reviewed the literature, and analyzed the various mechanisms proposed to explain the stability of nanodroplets and nanobubbles: interfacial mass transfer resistance, interfacial impurities, supersaturation, surface enhancement and dynamic equilibrium and contact line pinning. Unfortunately, as explained in detail in supplementary section S9, none of these mechanisms are able to explain our experimental results, and in particular, the growth of the islands. We have therefore, explored an alternative mechanism for island growth - the reduction in island solubility in the medium near the contact line arising from attractive disjoining stresses. The predictions of the island growth rate constants agree reasonably well with the experimental measurements. This is explained in detail in the manuscript on pages 8-12, and also covered in supplementary section S11.

- 2. The model system used in the manuscript is highly specific, because it contains no surfactant. Most emulsions contain surfactants to stabilize the emulsion drops. How would the mechanism work in the presence of surfactants that compete for the adsorption on the solid surface? This question has to be addressed.**

Response:

We thank the reviewer for this question. While surfactant-based systems were originally an area of future exploration for us, we have added some preliminary results to demonstrate the occurrence of this mechanism in the presence of surfactants in the suspending medium.

We performed experiments with a suspending phase of castor oil containing a nonionic surfactant, Span 80, at two different concentrations: 1.25 times CMC and 4 times CMC. (CMC value = 0.2 M^{-1}). The substrate used was plasma-treated SU8 and the drop phase was glycerol. Dramatic nucleation of glycerol islands was observed underneath the parent drop, with a foam-like structure between the islands (Movie 8 and Movie 9). The details of these experiments are provided on page 7 of the manuscript, and reproduced here:

“We were also curious as to whether this mechanism would occur in the presence of surfactants dispersed in the suspending phase, since most commercially available emulsion-based products are laden with interfacially-active chemicals. Surfactants are known to adsorb on solid surfaces and a possible outcome is that the nucleation of islands would be retarded, if not completely prevented⁴⁵⁻⁴⁸. However, from classical nucleation theory^{49,50}, surfactants are also known to reduce the interfacial tension of the liquid-liquid interface, and could, thus, accelerate the nucleation phenomenon. We conducted experiments with castor oil having pre-dissolved Span 80 at two different concentrations – 1.25 times the critical micelle concentration (CMC) and 4 times CMC (CMC value = 0.2 M^{-1}). Large, dense clusters of glycerol islands with foam-like features were observed underneath a glycerol parent drop approaching a plasma-treated SU8 substrate (Movie 8 and Movie 9), suggesting that the effect of reduction of the interfacial tension is likely a stronger effect for this system of oil, water, surfactant and substrate.”

In accordance with classical theory, the reduction in the interfacial tension between the drop and the suspending phase, can dramatically increase heterogeneous nucleation rates^{2,3}. In this case, it appears that the surfactant Span80 decreased the energy of the glycerol-silicone oil interface more strongly than the energy of the silicone oil-substrate interface, and hence the nucleation and growth of islands of glycerol is enhanced. However, if the substrate is covered with an interfacially-active species of stronger affinity that converts the substrate-oil interface to a low energy interface, but does not decrease the glycerol-oil interfacial energy substantially, then it is plausible that islands would not be observed. Our purpose in these preliminary tests was to demonstrate that the phenomenon discussed in this manuscript also happens in the presence of surfactants, and we have achieved that. Further work to understand the details of the effects of surfactants will be addressed in future work.

- 3. Throughout the manuscript, the authors argue that the van der Waals interaction is the dominating interaction. How can the authors exclude other interactions, e.g. electrostatics? Due to the larger difference in**

dielectric constants between the oil and the glycerin drops, charges might accumulate at the (nano) drop surface. The water content also will change the Debye screening length (compare discussion on page 6). This might give alternative interpretations of the effect of water content in the sample. To clarify these points data on the ion concentration in the sample is needed, e.g. dielectric spectroscopy or conductivity measurements.

Response:

We thank the reviewer for the question. We used the van der Waals interaction only to initiate our discussion of the interaction between the island and drop, and intended to convey that if only van der Waals interactions were considered, one would expect the island and the parent drop to eventually coalesce, as the drop and the island are composed of the same material. However, in our experiments involving silicone oil and glycerol, in many cases, we did not observe coalescence for several hours and sometimes even for days, so the conclusion presented was that there is a repulsive disjoining pressure acting to retard coalescence.

We are, however, grateful to the reviewer for raising this question as this has led us to explore a host of literature on charging of oil/water interfaces in the absence of electrolytes or charged surfactants. There are been a considerable amount of work done to show that a pristine water/hydrophobic interface (hydrophobic being oil or even air) can be negatively charged due to a selective adsorption of hydroxyl groups. The origin of free dangling hydroxyl groups can be attributed to the fact that interfacial water molecules have broken hydrogen bonds at a 2D surface such as a liquid-liquid interface rather than in the bulk^{4,5}.

Yaminsky et al. (2010) showed using a thin film balance setup that an air/water interface can be naturally charged with a charge potential of -57 mV with a Debye length of 152 nm⁶. The electric double layer (EDL) forces were sufficient to stabilize thick water films of the order of 100 nm. Addition of small amounts of electrolytes decreased the Debye length by screening this residual charge at the interface, causing film instability. Similarly, Ivanov and Kralchevsky (1997), Karraker and Radke (2002), Leunissen et al. (2007) and Creux et al. (2009), showed evidence of negative charge at an oil-water interface due to the difference in the dielectric constant between the two phases⁷⁻¹⁰. The zeta-potential was measured to be of $O(-100\text{ mV})$ with a strong pH dependence. The zeta potential dropped to zero for lower values of pH.

The zeta-potential at silicone oil-water interface was also measured by Gu and Li (1998) and reported to be -40 mV at neutral pH¹¹. No measurements were found

in the literature for silicone oil-glycerol interfaces, although we would not expect this interface to be charged, as glycerol cannot self-dissociate. Nevertheless, the glycerol we use does have trace amounts of water which could potentially charge the interface. It is not clear, though, whether water would preferentially arrange at a glycerol-silicone oil interface.

To test this, we conducted experiments with water drops settling in dehydrated SO1000 towards a plasma-treated SU8 substrate. Gu and Li (1998) paper showed that the isoelectric point for water-silicone oil interface, occurs at a pH of 5¹¹. In our latest experiments, we prepared pH 5 water by the addition of HCl. pH 5 water drops settling under gravity towards a SO1000-plasma-treated SU8 substrate, caused nucleation of water islands. Evidently, even at the isoelectric point, instant coalescence of the parent drop with the islands did not take place. Instead, the growth of the water islands, caused an upward motion of the parent drop, against gravity. This suggests that there is another source of the repulsive disjoining pressure, which we suspect to be due to the immobilization of silicone oil molecules at the oil-glycerol interface. We have also improved the discussion of this possibility in the supplementary section S7.

We have added this information into the main text (page 7):

“A curious observation in our experiments is that we witness extended growth periods (time scales ranging from several hours to days) of the nucleated islands that push the parent drop upwards, i.e. the islands and the parent drop do not coalesce readily. This is surprising, considering that coalescence between the parent drop and one of the islands is typically expected to be rapid for clean interfaces devoid of surfactants and in the presence of attractive van der Waals forces between like phases, indicating a repulsive barrier to coalescence. We rule out the possibility of electric double layer (EDL) forces caused due to interfacial charging at aqueous phase-silicone oil interfaces⁴⁴ by observing delayed coalescence even when experiments were conducted at the isoelectric point of the interface (pH 5) (see S7 and Movie 6). Furthermore, our experiments with glycerol-dehydrated castor oil combination did not show delayed coalescence (again, see S7), suggesting that the polymeric nature of the suspending medium plays a key role in the observed extended growth period of the islands. We can only speculate that the polymer chains are also immobilized at the SO-drop and SO-island interfaces. Based on previous work^{14,16,17}, we do know that the presence of small-molecule impurities such as dissolved water can disrupt the formation of an immobilised layer of polymer chains and thereby erase the oscillatory force behavior at reduced separations. We see that when silicone oil has trace amounts of dissolved water (~75% of the solubility of water in silicone oil, see S8; Movie 7), the parent drop merges with a nucleated site almost instantly upon contact. Also,

interferometric experiments wherein a glycerol drop approaches a flat glycerol-silicone oil (SO1000) interface show the formation of a steady film of silicone oil, whose film thickness is close to 10 nm (see S7). The reason for the presence of an immobilized polymer layer at the polymer-glycerol interface is not clear and needs further investigation, although we examine some possibilities in S7”

4. **On page 6, the authors argue about an immobilized layer on the nanodrops (“We can only speculate that ...”). To me this seems very speculative and needs a solid argument to support.**

Response:

As mentioned in the response to the previous comment, we have strengthened our discussion of this possibility by performing additional experiments, and by presenting additional arguments in supplementary section S7.

5. **Some major statement of the modelling part is unclear to me.**
- 5a. **The authors mention that diffusion is fast and not rate limiting. However, most of the arguments they give later on depend on diffusion (the boundary layer thickness, the growth of the nanodrop). Clarify.**

Response:

We thank the reviewer for this comment, and we apologize for the confusion created by our wording. We have now substantially improved our understanding and the modeling of the island growth. According to our measurements, diffusion can indeed be one of the rate determining steps in the wetting phenomenon. It can be a determinant not only of the starting instant of the phenomenon, but also of the growth of the island, as explained below.

Onset of island nucleation: We start our experiments with no glycerol in the suspending medium, and if the surface does not receive glycerol, islands cannot nucleate and grow. We need to determine the time instant at which the substrate first ‘sees’ glycerol, and this corresponds to the instant where the edge of the concentration boundary layer of glycerol that precedes the parent drop touches the surface. The concentration boundary layer thickness is a balance between diffusion and convection, and hence the diffusion phenomenon has to be included.

This allows us to set the time 't=0' for the new mechanism to take effect. According to our calculations, these thicknesses are on the order of a few microns.

Island growth: The arguments later in the paper focus on time scales after 't=0', when the drop gets closer to the surface, and there is an influx of glycerol into the substrate. We focus specifically on the limit when the total wetting time is dominated by the time for island growth caused by a diffusive addition of glycerol to the island and the bulk. This implies that the time required to nucleate islands, and the time required for bridge formation and coalescence between the drop and the island, are relatively negligible.

Our studies reveal two behaviours of island growth. For the silicone oil-glycerol-mica system, the island growth data suggests the trend, $b^2 = b_0^2 + k_c t$, which we reported in the original manuscript. We show in the current manuscript that this behaviour may be explained as being due to addition of glycerol to the island contact line controlled by diffusion across the immobilized polymer layer at the substrate. The constant k_c recovered from these arguments agrees well with the experimental measurements.

Also, after carefully analyzing our results for the experiments done with plasma treated SU8, we have discovered a second behaviour of island growth - $b^3 = b_0^3 + kt$. We show in the manuscript that this again may be explained by a diffusion-controlled addition of glycerol to the island via the contact line, but with concentration gradients established over the size of the island. The constant k calculated using these arguments again agrees well with experimental measurements.

The text from the manuscript that explains these ideas in detail is presented in pages 10 through 12 of the revised manuscript.

5b. Why is the volume growth rate of the drop (Eq. 2) proportional to the circumference of the nanodroplet?

Response:

As mentioned in our response to comment #1, our hypothesis is that the addition of glycerol to the island takes place due to a reduction in the solubility of glycerol near the contact line driven by an attractive disjoining pressure. When the island height becomes large, the addition of glycerol to the island is essentially via the

perimeter of the contact line, hence the proportionality to the circumference of the island. The constants corresponding to the island growth models estimated on the basis of this hypothesis agree well with the experimentally measured values.

- 5c. **The paragraph in the middle of page 9 (The experimentally measured critical height ...) is rather vague and unclear. Be more specific.**

Response:

We thank the reviewer for this comment. We have substantially reworked this paragraph. The critical height we assign is the minimum height of the RICM reconstructed film shape prior to film rupture. This has now been clarified in Fig. 5 caption.

The portion of Fig. 5 caption is reproduced below:

“The critical height, which is the film height at which the island intercepts the parent drop, as a function of the parent drop radius for glycerol drops of different sizes in castor oil and silicone oil settling towards plasma-treated SU8. The experimental critical heights are taken to be the minimum film height before the film ruptures (castor oil), or to be the film height prior to its increase due to island growth and delayed bridge formation (silicone oil).”

- 5d. **The used silicone oil or is poorly characterized? Molar mass distribution? Purity?**

Response:

Gel permeation chromatography (GPC)) results for silicone oil have now been reported in the main text under Materials and Methods section. The weight-averaged molecular weight (M_w) obtained from GPC for SO500 and SO1000 were 16 kDa and 20 kDa respectively, while the polydispersity index (PDI) was 1.73 and 1.49 respectively.

Proton Nuclear Magnetic Resonance (H-NMR) spectroscopy results of silicone oil are now presented in the supplementary section S12. No organic impurities were detected in the spectra; only peaks (~ 0.07 ppm) corresponding to hydrogen in silicone oil chains were observed.

- 5e. - On page 5 and 6 various contact angles are reported. Under which conditions are these contact angles measured? Drop phase? Continuum phase? Advancing or receding contact angle? Hysteresis?

Response:

The conditions at which the contact angles were measured have now been added in the main text. Also, a brief description of the measurement of the contact angles can be found in the ‘Experimental Methods’ section of the main text.

“It is instructive to assess the impacts of the type of substrate (in terms of wettability and surface roughness) and the materials composing the drop and suspending media, on the wetting mechanism. We examined spin-coated and cured SU8 substrates that were, after curing, used either directly (native SU8) or after plasma treatment in air (plasma-treated SU8). Native SU8, being a smooth substrate with low surface energy (SO1000-G-NS; static contact angle, $\theta \approx 89^\circ$) relative to mica and low roughness⁴⁰ (see S6), did not show evidence of nucleation (Movie 1). On the other hand, plasma treatment of SU8 is known to increase polar groups on the surface⁴⁰, and makes it preferentially wettable by glycerol (SO1000-G-PS; static contact angle, $\theta \approx 55^\circ$)”

“Static contact angles were measured by placing the substrate in a rectangular cuvette (10 x 10 x 45 mm, polystyrene cuvettes, Sarstedt AG & Co.). The cuvette was filled with the suspending medium and a large glycerol ($R \sim 300 \mu\text{m}$) drop was introduced using a silica capillary. The drop is allowed to settle towards the substrate and eventually spread on the substrate, after which images were taken using a camera and a 4X objective lens. These images were then used to measure the contact angle.”

6. - In the figures the font size varies strongly. The figures should be revised to have less than a factor of two between the largest and the smallest font (including sub- or superscripts).

Response:

We thank the reviewer for the suggestion. We have modified the figures to satisfy the reviewer’s requirement.

References:

1. Taylor, M. S. Stabilisation of water-in-oil emulsions to improve the emollient properties of lipstick. (University of Birmingham, 2011).
2. Lubetkin, S. & Blackwell, M. The nucleation of bubbles in supersaturated solutions. *Journal of Colloid and Interface Science* **126**, 610–615 (1988).
3. Chen, Q., Luo, L., Faraji, H., Feldberg, S. W. & White, H. S. Electrochemical Measurements of Single H₂ Nanobubble Nucleation and Stability at Pt Nanoelectrodes. *J. Phys. Chem. Lett.* **5**, 3539–3544 (2014).
4. Björneholm, O. *et al.* Water at Interfaces. *Chem. Rev.* **116**, 7698–7726 (2016).
5. Agmon, N. *et al.* Protons and Hydroxide Ions in Aqueous Systems. *Chem. Rev.* **116**, 7642–7672 (2016).
6. Yaminsky, V. V., Ohnishi, S., Vogler, E. A. & Horn, R. G. Stability of Aqueous Films between Bubbles. Part 1. The Effect of Speed on Bubble Coalescence in Purified Water and Simple Electrolyte Solutions. *Langmuir* **26**, 8061–8074 (2010).
7. Ivanov, I. B. & Kralchevsky, P. A. Stability of emulsions under equilibrium and dynamic conditions. *Colloids and Surfaces A: Physicochemical and Engineering Aspects* **128**, 155–175 (1997).
8. Karraker, K. A. & Radke, C. J. Disjoining pressures, zeta potentials and surface tensions of aqueous non-ionic surfactant/electrolyte solutions: theory and comparison to experiment. *Advances in Colloid and Interface Science* **96**, 231–264 (2002).
9. Leunissen, M. E., Blaaderen, A. van, Hollingsworth, A. D., Sullivan, M. T. & Chaikin, P. M. Electrostatics at the oil–water interface, stability, and order in emulsions and colloids. *PNAS* **104**, 2585–2590 (2007).
10. Creux, P., Lachaise, J., Graciaa, A., Beattie, J. K. & Djerdjev, A. M. Strong Specific Hydroxide Ion Binding at the Pristine Oil/Water and Air/Water Interfaces. *J. Phys. Chem. B* **113**, 14146–14150 (2009).
11. Gu, Y. & Li, D. The ζ -Potential of Silicone Oil Droplets Dispersed in Aqueous Solutions. *Journal of Colloid and Interface Science* **206**, 346–349 (1998).

RESPONSES TO THE COMMENTS AND QUESTIONS OF REVIEWER#2

1. I have found Fig. 2 to be a bit dense, and I think that the paper would gain from improvements here. First I would recommend that the notations be clarified. For example, \hat{h} is non-dimensionalized while \hat{R} is not. Moreover, the figure would benefit from a decrease in the information content of each subfigure. For example, provided notations are clarified, h_{ath} could be defined in Fig. 2b and its definition won't have to be repeated in 2d and 2f.
2. Legends should also be used to tighten the style of figures. For example, in Fig. 2b, it would be great to have a box stating that the black line is a fit of an exponential function to the data. The same holds for the other figures where fits are used.
3. Figure 2 is not self-consistent, i.e. reading the figure with its caption still leads to confusion and the need to refer either to the body of the text or, worse in my opinion, to the supplementary materials. I would suggest that the authors define all the quantities that are displayed in the figure (and their physical meaning) in the caption. For example, h_{trans} is not really well defined (and I find this true also in the text). I think that it is the thickness of the film for which the droplet switches from spherical to non-spherical. Is this correct? Can the authors clarify this? Finally, this leads me to not understand the difference between the data reported in 2b and 2d, reading the caption. Can the authors clarify this too? It seems to me that the distinction has to do with relative intensity of the lubrication and capillary pressure, but the way it does requires clarification.
4. I wonder about the dimensions of t^* in Fig. 2d. Given the equations in the figure, and especially the definition of h^{-1} , t^* has the dimensions of the inverse of a length, and therefore it is dimensionalized. I understand that this definition is used to test directly the dependence of the thickness on time, but the definition of t^* with no further commentary may somehow mislead the reader in thinking that this is a non-dimensionalized quantity. The same remark holds for \hat{t} in fig. 2f.

Combined response to 1, 2, 3 and 4:

We thank the reviewer for these questions, which have significantly improved the clarity of Figure 2. We have now made major changes to Figure 2 based on these suggestions. The rectifications are summarized below:

1. We have avoided the use of variables such as \hat{R} and instead presented the entire expression as the x and y labels. This has also allowed us to significantly reduce the amount information in each subfigure.
2. We attempted to add the legends, but the figure becomes extremely busy, so we have now provided more details in the caption.
3. The definitions of transition heights have now been clarified in the main text and are given below:

The first transition height denoting the regime change from $h \sim e^{-t}$ to $h^{-1} \sim t$, is given by $h_{trans_1} \approx \mathcal{A}(\Delta\rho g/\sigma)R^3$ where \mathcal{A} is measured to be 4.2 ± 0.6 . This transition occurs as the lubrication pressure starts to scale as the Laplace pressure while the film is still spherical.

A second transition height (h_{trans_2}) occurs for regime change from $h^{-1} \sim t$ to $h^{-3/2} \sim t$ where $h_{trans_2} \approx \mathcal{B}(\Delta\rho g/\sigma)R^3$, \mathcal{B} being measured to be 0.7 ± 0.1 . This transition occurs due to a change from a spherical to a dimpled film.

4. The definitions for t^* and \hat{t} had typos with missing terms for length scales. The definitions were supposed to be $t^* = c_2(\sigma h_{trans_1}/\mu R^2)t$ and $\hat{t} = c_3(\sigma h_{trans_2}^{3/2}/\mu R^{5/2}Bo^{1/2})t$ instead of $t^* = c_2(\sigma/\mu R^2)t$ and $\hat{t} = c_3(\sigma/\mu R^{5/2}Bo^{1/2})t$. These typos have now been corrected. Nevertheless, the data plotted were scaled correctly to render t^* and \hat{t} dimensionless.

Details of the scaling analysis are provided in Section S2.

5. **With respect to the fits used in Fig. 2, the authors say nothing about the prefactors that they found. Are these prefactors of order 1?**

Response:

We have now provided a table of the various prefactors appearing in figure 2 in a separate table in the supplementary information (S2). We make a mention of this table in the caption of Figure 2. The prefactor c_1 and c_3 are indeed of order unity. However, in the regime where the drop retains its spherical shape but with a film pressure that scales as the Laplace pressure, the constant c_2 is of the order of 0.1.

6. **The authors suggest that nucleation should start once the oil film thickness is comparable to the thickness of the glycerol boundary layer in**

the vicinity of the droplet. I might have missed the test of this claim in the paper. Can the authors discuss this?

Response:

We did not directly test the idea experimentally, but we inferred it from our experimental observations. We know that the silicone oil is initially devoid of glycerol. We also know that the islands are observed to nucleate and grow only when the drop is in the vicinity of surface. When the glycerol drop is very far away from the surface and descending towards the surface, the dissolved glycerol is restricted to lie in the vicinity of the drop in a thin concentration boundary layer governed by the balance of convection and diffusion; the surface simply does not ‘see’ the glycerol at this point. As the drop approaches the surface, the edge of the concentration boundary layer preceding the drop touches the surface, only upon which the surface begins to receive glycerol. We have used an estimate of this film thickness to predict the critical height for coalescence between the island and the parent drop, and the agreement with experiments is reasonably good (see Fig. 5).

In the cases where mica is used as the substrate, the rate limiting step for glycerol transport is diffusion through the immobilized polymer layer (see page 11 of manuscript). While the boundary layer thickness is of the order of a few microns, the slow down in mass transfer through a ten nm thick immobilized silicone oil layer can cause significant delay in the onset of nucleation.

7. **In the supplementary materials, I am not sure about the prefactors in eq. S2.7. My own calculation gives me:**

$$-\partial_t h \approx \frac{h}{a} \left[\frac{Gh^2}{12\mu} + \frac{U}{2} \right]$$

Directly integrating this equation with the condition that $h(t=0) = h_0$ leads me to an exponential with a different argument. Where does the difference lie? Yiantsios and Davis proceed with this equation by applying asymptotics to it. Did the authors do this too? Note that integrating the equation above with the definitions of P and a given in Table 1 leads me to the same scalings as the authors.

Response:

This is a typographical error in the velocity used in the text. The equation the reviewer has provided is correct.

$$-\partial_t h \approx \frac{h}{a} \left[\frac{Gh^2}{12\mu} + \frac{U}{2} \right]$$

But we note that this does not change any of the comparisons in the manuscript, because the theories are based on scaling, and the above equation has a scaling based addition on the right hand side (there are order unity prefactors in front of each term in the bracket).

As requested by the reviewer, we have now provided values of the various prefactor of the comparisons with the scaling theory. We have also refined our arguments leading up to the scaling relationships in S2, as explained in the next response.

8. **Besides this discrepancy, I am able to find the solutions of the authors only if I assume that the Couette part of the flow is weak compared to the Poiseuille part. This assumption is not necessarily obvious. Can the authors comment on this? My understanding, given the systems they use is that they have a viscosity ratio of order 1, and it is not obvious to me that the Couette component can be neglected. A few words about this, maybe referring to Yiantsios and Davis who discuss the impact of the viscosity ratio, may be useful and instructive to the reader.**

Response:

The velocity profile in the thin film can vary between the case of a pure Poiseuille flow for large viscosity ratio [see subfigure (a) below], to the case of a Couette-Poiseuille flow with a stress free boundary condition for low viscosity ratios [see subfigure (b) below].

In general, we can write

$$u \sim u_t + u_p$$

where u_p is the Poiseuille contribution and u_t is the Couette contribution. A tangential stress balance at the interface yields

$$\frac{\hat{\mu}u_t}{a} \sim \frac{\mu(u_p - u_t)}{h},$$

which can be rearranged to obtain the following relationship between u_t and u_p .

$$u_t \sim \frac{m}{(m+1)}u_p$$

where m is the mobility ratio, defined as

$$m = \frac{a}{\lambda h} = \frac{1}{\lambda \varepsilon}$$

with $\varepsilon = h/a$, to distinguish between immobile and fully mobile interfaces. This ratio can range from very small values to very large values, depending on the viscosity ratio, the film size and the thin film height. For $m \ll 1$, the Couette contribution is much weaker than the Poiseuille contribution ($u_t \ll u_p$) while for $m \gg 1$, the Couette and Poiseuille contributions are comparable ($u_t \sim u_p$). The total flow in the film is, thus,

$$u \sim \left(\frac{m}{m+1} + 1 \right) u_p.$$

It can be seen that while the viscosity ratio (through m) affects the actual magnitude of the total efflux velocity u , it does not affect the *scaling* of u , i.e. $u \sim u_p$ for both $m \ll 1$ and $m \gg 1$. In all of our scaling comparisons, the viscosity ratio was moderate (0.8 to 5) and lay in the limit $\varepsilon \ll \lambda \ll 1/\varepsilon$ identified by Yiantsios and Davis¹, i.e. we were always in the fully mobile limit in all the recorded data of h vs t . Therefore, even if the transition from an immobile to mobile regime occurred (which would lead to a dependence on the viscosity ratio), it had already manifested prior to the recording of the h vs t data. This is why the drop viscosity does not feature in the scaling of the minimum height evolution with time.

This discussion has now been appended to supplementary section S2. We note that the rate of drainage of a film between a solid surface and a drop can be significantly different from the film between two drops of the same material. In the latter case, the Couette component can be significantly larger than the Poiseuille contribution (e.g. see Ramachandran and Leal, Phys. Rev. Fluids, 2016²). This does not happen in the flow in the film between a drop and a rigid surface.

References

1. Yiantsios, S. G. & Davis, R. H. On the buoyancy-driven motion of a drop towards a rigid surface or a deformable interface. *Journal of Fluid Mechanics* **217**, 547–573 (1990).
2. Ramachandran, A. & Leal, L. G. Effect of interfacial slip on the thin film drainage time for two equal-sized, surfactant-free drops undergoing a head-on collision: A scaling analysis. *Phys. Rev. Fluids* **1**, 064204 (2016).

RESPONSES TO THE COMMENTS AND QUESTIONS OF REVIEWER#3

We are grateful to reviewer for the careful perusal of our manuscript and the critical comments, which have motivated us to improve the manuscript significantly. We have provided a point by point response to the reviewer's comments below:

- 1. The diffusion of liquid from drops into an immiscible fluid and its impact on its surrounding has been shown many times in different applications/fields such as microfluidics, water evaporation and emulsions. The authors can have also a look at the following papers: Su et al, Langmuir 2013,29,13339, Shimizu &Tanaka , ncomms 6,2015, 7406, Carrier et al. J. Fluid Mechanics, 2016, 798.**

Response:

We respectfully disagree with the reviewer's comment. It is obvious that diffusion of liquid from drops is not new, and we also do not deny the existence of literature that talks about coarsening of emulsion droplets when surrounded by a seemingly immiscible system. But this is not the novelty claimed in the manuscript. The novelty is the fact that the drop can wet a surface by a mechanism that involves coalescence with islands on the surface that originate from the drop fluid itself. We claim that even a small solubility of the drop fluid in the medium can lead to this wetting mechanism, if the substrate material and texture are conducive to the nucleation of the drop fluid. We further enumerate the new findings of our work below:

1. Small islands grow at the expense of the large drop, which is the opposite of what one would expect by Ostwald ripening.
2. Our manuscript provides clues to the solution of the long standing question of the unexpectedly long stability of nanobubbles on surfaces¹. We provide a hypothesis based on a reduced solubility near the contact line due to attractive forces between the drop phase and the solid substrate. Our scaling estimates of island growth rates based on this hypothesis agree with our measurements.
3. Wetting driven by coalescence leading to new parameters that can be used to control wetting rates, such as solubility, diffusivity, surface coatings, etc.
4. If the mechanism is coalescence-based, the wetting times can be significantly shorter than those predicted by current theories based on direct drop to surface bridging and wetting.

5. A supersaturation of the drop fluid in the bulk solution is not required to see the nucleation phenomenon.

The articles cited by the reviewer explain: a) predictive models for dissolution of single emulsion droplets, b) hydrodynamically driven, high volume fraction emulsion drop coarsening, and c) evaporation and shielding effects of an array of drops, respectively. These articles do not in any way imply or suggest the mechanism that we have discussed in our manuscript.

- 2a. **The authors pretend that the results can give insights for the design of emulsion based coating materials. However the system studied in the manuscript is surfactant free .**

Response:

We thank the reviewer for this question. While surfactant-based systems were originally an area of future exploration for us, we have added some preliminary results to demonstrate the occurrence of this mechanism in the presence of surfactants in the suspending medium.

We performed experiments with a suspending phase of castor oil containing a nonionic surfactant, Span 80, at two different concentrations: 1.25 times CMC and 4 times CMC (CMC value = 0.2 M^{-2}) . The substrate used was plasma-treated SU8 and the drop phase was glycerol. Dramatic nucleation of glycerol islands underneath the parent drop was observed, with a foam-like structure between the islands (Movie 8 and Movie 9). The following discussion as been appended to the manuscript on page 7:

“We were also curious as to whether this mechanism would occur in the presence of surfactants dispersed in the suspending phase, since most commercially available emulsion-based products are laden with interfacially-active chemicals. Surfactants are known to adsorb on solid surfaces and a possible outcome is that the nucleation of islands would be retarded, if not completely prevented⁴⁵⁻⁴⁸. However, from classical nucleation theory^{49,50}, surfactants are also known to reduce the interfacial tension of the liquid-liquid interface, and could, thus, accelerate the nucleation phenomenon. We conducted experiments with castor oil having pre-dissolved Span 80 at two different concentrations – 1.25 times the critical micelle concentration (CMC) and 4 times CMC (CMC value = 0.2 M^{-1}). Large, dense clusters of glycerol islands with foam-like features were observed underneath a glycerol parent drop approaching a plasma-treated SU8 substrate

(Movie 8 and Movie 9), suggesting that the effect of reduction of the interfacial tension is likely a stronger effect for this system of oil, water, surfactant and substrate.”

In accordance with classical theory, the reduction in the interfacial tension between the drop and the suspending phase, can dramatically increase heterogeneous nucleation rates^{3,4}. In this case, it appears that the surfactant span-80 decreased the energy of the glycerol-silicone oil interface more strongly than the energy of the silicone oil-substrate interface, and hence the nucleation and growth of islands of glycerol is enhanced. However, if the substrate is covered with an interfacially-active species of stronger affinity that converts the substrate-oil interface to a low energy interface, but does not decrease the glycerol-oil interfacial energy substantially, then it is plausible that islands would not be observed. Our purpose in these preliminary tests was to demonstrate that the phenomenon discussed in this manuscript also happens in the presence of surfactants, and we have achieved that. Further work to understand the details of the effects of surfactants will be addressed in future work.

- 2b. Moreover the size of their drops is much larger (100 microns) than what one usually finds in emulsions (typically micron size). This will change the capillary pressure in the drops by a factor of 100, and can consequently change all the behaviour of the system.**

Response:

We thank the reviewer for this comment. It would have been useful if the reviewer had mentioned the specific effect or effects due to capillary pressure that the reviewer thinks would affect the behaviours of the system relevant to the mechanism discovered here. We outline below as many effects of drop size we can think of:

- a) As mentioned by the reviewer, smaller drops lead to higher capillary pressures. But they would also result in greater solubility of the drop fluid in the medium. The larger the solubility, the greater the driving concentration difference for mass transfer to the surface and the island, and the lower the loss rate from the island back to the drop.
- b) When the drops are only a few microns in size, the hydrodynamic drainage period is predominantly in the spherical drop regime of drainage. A dimpled phase of film drainage is typically not observed for small drops owing to the

large lubrication pressures required for interface deformation. If predictions of wetting time are made with current theories based on drops-substrate disjoining pressure interactions and bridge formation, the trend of the drainage time with drop radius would be *opposite* to what is measured by us and expected from the coalescence-based theory in our manuscript. (See pages 12-13 of manuscript).

- c) Micron and submicron drops would lead to Brownian forces becoming important. We determine the scale of the drop radius R_c below which Brownian forces are stronger than gravitational forces by equating $\Delta\rho g(4\pi R_c^3/3) = KT/R_c$, which yields $R_c = (3KT/4\pi\Delta\rho g)^{1/4}$. Substituting typical values of $KT = 4 \times 10^{-21}$ J, $\Delta\rho \sim 100$ kg/m³, and $g = 9.8$ m/s², we get $R_c \sim 1$ μ m. For $R \ll R_c$, Brownian forces dominate and the appropriate definition of a ‘Bond’ number (as used in the manuscript) is $Bo = KT/(\sigma R^2)$. For $R \gg R_c$, gravitational forces are stronger, and the Bond number definition used in the manuscript is more appropriate. We can estimate the capillary number corresponding to Brownian forces pushing a drop towards the surface. Here KT is the thermal energy, σ is the interfacial tension and R is the drop radius. For drops of radii $R = 100$ nm to 1 μ m, and $\sigma \sim 10$ mN/m, the Bond numbers range between 10^{-5} to 10^{-7} , i.e. they are weak, implying that sub-micron drops are likely to drain predominantly in the spherical regime. As mentioned in (c), our coalescence based theory predicts a drainage time trend with R opposite to traditional film drainage theories.

According to (a), (b) and (c), smaller drop sizes would, in fact, make the coalescence mechanism of wetting more important to consider.

- 3. The authors claim that immobilization of the polymeric film between the substrate and drop leads to a highly viscous film with reduced degrees of freedom for the chains that can be structured parallel to the substrate. If this is the case I suppose that locally the diffusion coefficient should become very small especially if we will be in the presence of an anisotropic film. So, why then will diffusion be faster in this regime compared to the unconfined surrounding of the drop, in a way to induce the ‘colonization’? In a similar situation, confined crystal growth experiments close to a substrate separated by a thin film of solution (Royne et al., J.Cryst.Growth ,86, 349, 2012, Cagliardi,**

Phys.Rev E, 97,012802 2018), it has been shown that the diffusion of ions in this thin film is rather very slow compared to the bulk and consequently this leads to some peculiar crystal shapes. Such arguments should also be valid in the experiments described here.

Response:

We thank the reviewer for this thought-provoking question. The reviewer is correct in raising the argument that diffusion can be significantly slowed down due to polymer confinement. Our substantially improved mathematical model now incorporates this aspect and the experimentally measured island growth rates can only be rationalized when accounting for the reduced rates of mass transfer through this immobilized layer.

The immobilization of polymer chains occurs mainly due to adsorption and pinning of the chains to the substrate⁵. Hence, this layer with a different conformation, will be found *everywhere* on the substrate, irrespective of whether it is sandwiched between a drop and the substrate. The diffusing molecules of the drop phase will then encounter this layer not only underneath the drop, but also at other portions of the substrate. In order for islands to nucleate and grow, mass transfer must now occur over this barrier with significantly lowered diffusion coefficients.

Two types of growth behaviors are observed for the two types of substrates used—mica and plasma-treated SU8, when silicone oil was used as the suspending medium. For glycerol islands growing in silicone oil, a power law of $b^2 \sim t$ is observed on a mica substrate and $b^3 \sim t$ is observed on a plasma-treated substrate. Both scaling behaviors have the same underlying mechanism of addition of glycerol to the contact line of an island due to reduced solubility. However, the rate-limiting step for mass transfer is different. As pointed out by the reviewer, the diffusivity of glycerol in the immobilized layer is significantly lowered from that in the bulk. We have deduced the diffusivity in the immobilized film to be $D \sim 10^{-15} \text{ m}^2/\text{s}$ as opposed to $D \sim 10^{-11} \text{ m}^2/\text{s}$ in bulk silicone oil (see S14 and pages 11-12 of manuscript).

The reduced diffusivity in the immobilized silicone oil layer on mica substrates was also the reason why we did not construct a plot of critical height h_c vs drop radius R for this case in Fig. 5 of the revised manuscript. While the boundary layer thickness δ can be a few microns thick, the appreciable reduction in mass transfer

through the immobilized silicone oil layer can lead to significant delay in the onset of nucleation of islands.

We now explain why islands are observed over a certain length scale from the center of the contact zone formed between the drop and the surface. In majority of our experiments, the suspending medium is devoid of the drop fluid at the outset. When the drop is close to the surface, a front of dissolved glycerol grows around the drop and over the substrate. Therefore, in the vicinity of the drop, the dissolved glycerol concentration is close to saturation, whereas far away from the drop, the medium is depleted in glycerol. As shown in S11, for the manifestation of the islands, it is essential that the concentration of dissolved drop phase around an island be close to the saturation concentration. A lower concentration of dissolved drop phase would compromise the stability of a nucleated island causing the island to dissolve into the bulk due to Ostwald ripening based on the Kelvin equation. Hence, in our experiments, islands are found on the substrate only under the drop and close to the drop.

4. **Figure 3c shows larger islands at the extremities of the parent drop compared to the centre whereas figures 3b and 4 and the movies show a more homogeneous distribution . Is there any explanation for this observation ?**

Response:

We thank the reviewer for this question. It helped us realize that figure 3c can be confusing. The short answer to the question is that in figure 3c, the axis of symmetry of the parent drop is located on the left edge of the SEM image while the drop periphery is situated towards the right edge. The larger islands are present near the center (left edge of the image) and the island size progressively decreases as we move towards the drop periphery (right edge of the image).

To better understand the cryo-SEM image, we refer to the figure panels R1 and R2 below. The protocol developed was as follows:

- 1) Glycerol drops were allowed to settle under gravity in PDMS [Sylgard 184 + cross-linker (10:1 ratio)] towards a freshly cleaved mica surface (Figure R1 a). The PDMS was then allowed to cure slowly under ambient conditions for 12 hours.
- 2) The cured sample with embedded glycerol drops was flipped upside down and stuck on an SEM stub with electrically conductive epoxy. (Figure R1 b).

- 3) Using a Sharpie®, the locations of glycerol drops were marked while simultaneously observing under a stereomicroscope (Figure R2 a).
- 4) Under SEM imaging, a trench was milled using Focused Ion Beam milling near the center of the Sharpie marking (Figure R2 b, c, d, e).

Note that the Figure R2 e is the same as Figure 3c, but rotated by 180°. Figure R2 f is the stereomicroscope image of the same sample after milling the trench. We can then see that the right edge of Figure R2e (or left edge of Figure 3c) is close to the drop axis of symmetry. The islands near the axis of symmetry are larger while the islands away are smaller.

Figure R1.

Figure R2.

We have now included a sentence in the caption of Figure 3c that specifies the locations of the axis and periphery of the drop.

5. **Is there any critical length for the ‘nucleation’ of the islands? Again in classical nucleation and growth such a critical length is present (see for example J. Aizenberg et al, Nature 398, 1999).**

Response:

The journal paper referred to by the reviewer details selective growth of calcite crystals on polar SAM groups patterned with a background of methyl-terminated SAM on a substrate. The methyl-terminated regions were devoid of crystal growth by ensuring that the polar SAM patches were patterned at a distance less than a critical length wherein undersaturation prevails.

In this work, we stress that supersaturation is unnecessary for nucleation to occur. A reduction in solubility near the contact line due to attractive forces between the nucleating phase and the substrate is sufficient for islands to form, provided the concentration of the dissolved phase is close to saturation levels. In this case, there is a region of depletion whose length scale is b in the absence of the immobilized polymer, and d_{im} in its presence. We refer the reviewer to pages 9 to 12 of the revised manuscript for the discussion of the details of the mass transfer process.

6. **In addition, it could well be that first a microscopic film forms on the surface, which subsequently destabilizes into the droplet configuration.**

Response:

We thank the reviewer for this thought. Although we expect an augmentation of the concentration of the drop fluid near the surface due to its affinity to the substrate, we cannot confirm the presence of such a nanoscopic layer. If the film thickness were significant (greater than a few nanometers), and had at least a micron scale lateral size, we would have detected the film with the RICM technique. But we did not see any evidence of this. This, however, does not affect the analysis in the manuscript, which considers the addition of glycerol via diffusion from the bulk to the contact line to be the rate limiting step. The chemical potential of the glycerol in the surface enriched layer is the same as the bulk chemical potential of glycerol at equilibrium^{6,7}, and since mass transfer is a result of chemical potential gradients, the glycerol molecules in the vicinity of the substrate are expected to have the same mass transfer properties as the bulk dissolved glycerol.

References

1. Lohse, D. & Zhang, X. Surface nanobubbles and nanodroplets. *Rev. Mod. Phys.* **87**, 981–1035 (2015).
2. Taylor, M. S. Stabilisation of water-in-oil emulsions to improve the emollient properties of lipstick. (University of Birmingham, 2011).
3. Lubetkin, S. & Blackwell, M. The nucleation of bubbles in supersaturated solutions. *Journal of Colloid and Interface Science* **126**, 610–615 (1988).
4. Chen, Q., Luo, L., Faraji, H., Feldberg, S. W. & White, H. S. Electrochemical Measurements of Single H₂ Nanobubble Nucleation and Stability at Pt Nanoelectrodes. *J. Phys. Chem. Lett.* **5**, 3539–3544 (2014).
5. Horn, R. G. & Israelachvili, J. N. Molecular organization and viscosity of a thin film of molten polymer between two surfaces as probed by force measurements. *Macromolecules* **21**, 2836–2841 (1988).
6. Brenner, M. P. & Lohse, D. Dynamic Equilibrium Mechanism for Surface Nanobubble Stabilization. *Phys. Rev. Lett.* **101**, 214505 (2008).
7. Petsev, N. D., Shell, M. S. & Leal, L. G. Dynamic equilibrium explanation for nanobubbles' unusual temperature and saturation dependence. *Phys. Rev. E Stat. Nonlin. Soft Matter Phys.* **88**, 010402 (2013).

RESPONSES TO THE COMMENTS AND QUESTIONS OF REVIEWER#4

“It indeed a very good work that deserve to be published in Nature Comm. I have only one concern which is about the uncertainties that are not provided on Figs. 2c, 2e.”

Response:

We thank the reviewer for this. We have now added uncertainties to the figures 2c, e and f.

REVIEWERS' COMMENTS

Reviewer #2 (Remarks to the Author):

Here is my review for the revised version of this paper.

The authors have addressed my initial comments successfully. The additions they have made to the paper to answer the other reviewers' questions make the paper even stronger. This paper can be published.

I have only a very minor comment pertaining to the equations written in Figs 2d and g. The figure would be clearer if the authors wrote the equations in the form $h/h_{trans,1} = (1 + \frac{c_2 \sigma}{h_{trans,1}}) \mu R^2 t^{-1}$.

Reviewer #5 (Remarks to the Author):

This manuscript is of very high quality, and I recommend publication, basically as is.

The authors propose a new mechanism for droplets wetting surfaces. The standard picture involves a droplet approaching a surface, progressively squeezing out the film of the external phase until non-hydrodynamic forces (typically van der Waals / disjoining pressure) drive the disruption of the draining film. In this manuscript, the authors argue that another mechanism can occur in some liquid/solid systems — namely, that molecules of the dispersed-phase liquid may diffuse through the film and nucleate droplets on the solid, whereupon the 'wetting' event actually occurs via a coalescence between the new droplets, rather than a direct wetting. The authors present compelling experimental, scaling, and theoretical evidence to support their proposal, along with control experiments that reproduce the classical theory when expected. Given the ubiquity of wetting processes in science, industry and technology, and the complexity of the underlying dynamical process, I feel this manuscript represents an extremely important development that is both provocative and well-supported.

The authors' response to the referee's first reports were truly impressive. Several referees raised questions that were deep, detailed and on point; in those cases, the authors engaged those issues in a serious and thoughtful way, and clarified their manuscript, made adjustments, or performed and described additional experiments. One referee was more negative, but (in my view) did not provide much support for his/her assertions. In some cases, those critiques were simply off base and seemed based upon a list of references that bore only superficial resemblance to the present work. In other cases, the referee raised issues that I felt would not necessarily be appropriate because of key differences in between the referee's example and the present physical system. I was impressed that the authors engaged these ideas deeply as well, and found ways to improve their manuscript and to address additional ideas.

In short, I think this manuscript makes a convincing experimental and theoretical case for a new mechanism in substrate wetting, which is a process of substantial importance that has been studied for many decades, yet remains still elusive. The idea was clear and compelling, and the evidence was powerful and convincing.

Reviewer #6 (Remarks to the Author):

I recommend the publication of the manuscript, with a few additions/ suggestions. The experimental results included here are both original and inspire deep curiosity. The discussion is to the point and presents a great starting point for future research in this area.

My review in my capacity as an adjudicator is primarily aimed at addressing the very detailed exchange between the authors and the reviewers. I think the authors have made a great effort to address every comment and suggestion. I have not seen the original manuscript that was submitted, but the revised version reads quite well.

There are a few additional suggestions.

(a) The schematics show islands, but without comment on the height and the aspect ratio of droplets that emerge on the substrate. This can be mentioned explicitly in Figure captions and in the text.

(b) The author's mentioned the mechanism proposed by Vrij for spontaneous rupture of foam and supported films. The authors might as well include Yilixiati et al, *Molecular Systems Design & Engineering*, 2019 that shows the only example of this mechanism in freestanding films.

(c) Formation of islands and their shape, growth & coalescence is indeed detailed quite well in Zhang and Lohse review from 2015, but the authors might find their 2018 *Soft Matter* paper and 2020 review as additional references. Likewise, Zhang, Yilixiati, Pearsall & Sharma *ACS Nano*, 2016 & *Langmuir* 2018 show formation of nanoscopic mesas in foam films with diameter in microns and height in nm, impacted by disjoining pressure. The papers could also be mentioned for including for completeness a mention of the fact that disjoining pressure itself changes due to additional contributions for surfactant solutions, especially above CMC.

(d) The authors might find it useful to mention spreading studies by Walls, Haward, Shen and Fuller, *Physical Review Fluids* 2016, as a contrasting mechanism.

RESPONSES TO REVIEWER #2

1. **The authors have addressed my initial comments successfully. The additions they have made to the paper to answer the other reviewers' questions make the paper even stronger. This paper can be published.**

Response: We thank the reviewer for the comment.

2. **I have only a very minor comment pertaining to the equations written in Figs 2d and g. The figure would be clearer if the authors wrote the equations in the form**
$$h/h_{\text{trans},1} = (1 + \frac{c_2 \sigma h_{\text{trans},1}}{\mu R^2 t})^{-1}.$$

Response: We have now corrected this.

RESPONSES TO REVIEWER #5

This manuscript is of very high quality, and I recommend publication, basically as is.

The authors propose a new mechanism for droplets wetting surfaces. The standard picture involves a droplet approaching a surface, progressively squeezing out the film of the external phase until non-hydrodynamic forces (typically van der Waals / disjoining pressure) drive the disruption of the draining film. In this manuscript, the authors argue that another mechanism can occur in some liquid/solid systems – namely, that molecules of the dispersed-phase liquid may diffuse through the film and nucleate droplets on the solid, whereupon the ‘wetting’ event actually occurs via a coalescence between the new droplets, rather than a direct wetting. The authors present compelling experimental, scaling, and theoretical evidence to support their proposal, along with control experiments that reproduce the classical theory when expected. Given the ubiquity of wetting processes in science, industry and technology, and the complexity of the underlying dynamical process, I feel this manuscript represents an extremely important development that is both provocative and well-supported.

The authors’ response to the referee’s first reports were truly impressive. Several referees raised questions that were deep, detailed and on point; in those cases, the authors engaged those issues in a serious and thoughtful way, and clarified their manuscript, made adjustments, or performed and described additional experiments. One referee was more negative, but (in my view) did not provide much support for his/her assertions. In some cases, those critiques were simply off base and seemed based upon a list of references that bore only superficial resemblance to the present work. In other cases, the referee raised issues that I felt would not necessarily be appropriate because of key differences in between the referee’s example and the present physical system. I was impressed that the authors engaged these ideas deeply as well, and found ways to improve their manuscript and to address additional ideas.

In short, I think this manuscript makes a convincing experimental and theoretical case for a new mechanism in substrate wetting, which is a process of substantial importance that has been studied for many decades, yet remains still elusive. The idea was clear and compelling, and the evidence was powerful and convincing.

Response: We thank the reviewer for the encouraging remarks.

RESPONSES TO REVIEWER #6

- 1. I recommend the publication of the manuscript, with a few additions/suggestions. The experimental results included here are both original and inspire deep curiosity. The discussion is to the point and presents a great starting point for future research in this area.**

My review in my capacity as an adjudicator is primarily aimed at addressing the very detailed exchange between the authors and the reviewers. I think the authors have made a great effort to address every comment and suggestion. I have not seen the original manuscript that was submitted, but the revised version reads quite well..

Response: We thank the reviewer for the response

- 2. The schematics show islands, but without comment on the height and the aspect ratio of droplets that emerge on the substrate. This can be mentioned explicitly in Figure captions and in the text.**

Response: We have now indicated that the figure is not to scale (the islands are much smaller than the parent drop), and that the aspect ratio of the islands in most of our experiments is large ($b \gg l$). The following sentences have been added to Fig. 1c caption:

“Note that the schematic shown is not drawn to scale. The islands are much smaller than the parent drop, and the aspect ratio depends on the island contact angle”

- 3. The author's mentioned the mechanism proposed by Vrij for spontaneous rupture of foam and supported films. The authors might as well include Yilixiati et al, Molecular Systems Design & Engineering, 2019 that shows the only example of this mechanism in freestanding films.**

Response: We have now cited the article Yilixiati et al, Molecular Systems Design & Engineering, 2019 (Reference# 9)

- 4. Formation of islands and their shape, growth & coalescence is indeed detailed quite well in Zhang and Lohse review from 2015, but the authors might find their 2018 Soft Matter paper and 2020 review as additional references.**

Response: We have now cited both the articles.

- 5. Likewise, Zhang, Yilixiati, Pearsall & Sharma ACS Nano, 2016 & Langmuir 2018 show formation of nanoscopic mesas in foam films with diameter in microns and height in nm, impacted by disjoining pressure. The papers could also be mentioned for including for completeness a mention of the fact that disjoining pressure itself changes due to additional contributions for surfactant solutions, especially above CMC.**

Response: We have shown only preliminary experiments with surfactant-based systems and hence did not present the effect of micellar structuring leading to stratification of thin films during drainage. These papers will be cited in a future publication that is currently being prepared in the group.

- 6. The authors might find it useful to mention spreading studies by Walls, Haward, Shen and Fuller, Physical Review Fluids 2016, as a contrasting mechanism**

Response: Walls, Haward, Shen and Fuller, PRF, 2016: This work deals with the advancement of the contact line during the spreading of a sessile drop while immersed in a miscible liquid medium. While our work deals with the mechanism prior to spreading or rather before the instability of the suspending medium film, this paper deals with the physics of spreading after film rupture has occurred, and is unrelated to our paper. Hence, we have chosen to not cite this work.